# The Arabidopsis V-ATPase is localized to the TGN/EE via a seed plant-specific motif

Upendo Lupanga[1], Rachel Röhrich[1], Jana Askani[1], Stefan Hilmer[2], Christiane Kiefer[3], Melanie Krebs[1], Takehiko Kanazawa[4,5], Takashi Ueda[4,5], Karin Schumacher[1]*

[1]Department of Cell Biology, Centre for Organismal Studies, Heidelberg University, Heidelberg, Germany; [2]Electron Microscopy Core Facility, Heidelberg University, Heidelberg, Germany; [3]Department of Biodiversity and Plant Systematics, Centre for Organismal Studies, Heidelberg University, Heidelberg, Germany; [4]Division of Cellular Dynamics, National Institute for Basic Biology, OkazakiAichi, Japan; [5]The Department of Basic Biology, SOKENDAI (The Graduate University for Advanced Studies), OkazakiAichi, Japan

**Abstract** The V-ATPase is a versatile proton-pump found in a range of endomembrane compartments yet the mechanisms governing its differential targeting remain to be determined. In Arabidopsis, VHA-a1 targets the V-ATPase to the TGN/EE whereas VHA-a2 and VHA-a3 are localized to the tonoplast. We report here that the VHA-a1 targeting domain serves as both an ER-exit and as a TGN/EE-retention motif and is conserved among seed plants. In contrast, Marchantia encodes a single VHA-isoform that localizes to the TGN/EE and the tonoplast in Arabidopsis. Analysis of CRISPR/Cas9 generated null alleles revealed that VHA-a1 has an essential function for male gametophyte development but acts redundantly with the tonoplast isoforms during vegetative growth. We propose that in the absence of VHA-a1, VHA-a3 is partially re-routed to the TGN/EE. Our findings contribute to understanding the evolutionary origin of V-ATPase targeting and provide a striking example that differential localization does not preclude functional redundancy.

*For correspondence: karin.schumacher@cos.uni-heidelberg.de

Competing interests: The authors declare that no competing interests exist.

## Introduction

Compartmentalization into distinct membrane-bound organelles that offer different chemical environments optimized for the biological processes that occur within them is a hallmark of eukaryotic cells. Vacuolar-type $H^+$-ATPases (V-ATPases) are rotary engines that couple the energy released by ATP hydrolysis to the transport of protons across membranes of several intracellular compartments. They consist of two subcomplexes: The cytosolic $V_1$ subcomplex responsible for ATP hydrolysis composed of eight different subunits (A, B, C, D, E, F, G, H) and the membrane-integral $V_O$ subcomplex consisting of 6 subunits (a, d, e, c, c', c') required for proton translocation (*Sze et al., 2002*). Although all eukaryotic V-ATPases are strikingly similar regarding their structure and biochemical activity, their biological reach has been greatly diversified by cell-type specific expression and differential subcellular localization (*Cotter et al., 2015*). Differential targeting of the V-ATPase is mediated by isoforms of subunit a, the largest of the V-ATPase subunits that consists of a C- terminal hydrophobic domain with eight transmembrane domains and a large N-terminal domain that is accessible for cytosolic interaction partners (*Zhao et al., 2015*; *Vasanthakumar et al., 2019*). Previous studies in yeast and plants have shown that the targeting information is contained in the N-terminal domain (*Dettmer et al., 2006*; *Kawasaki-Nishi et al., 2001a*), however the responsible targeting domain has so far only been addressed for the yeast isoforms Vhp1p and Stv1p. Whereas Vph1p targets the yeast V-ATPase to the vacuole, a motif containing an aromatic residue (WKY) in

the N-terminus of Stv1p targets the V-ATPase to the Golgi/endosomal network (*Finnigan et al., 2012*). The localization of Stv1p at the Golgi is dependent on interaction of the WKY motif with phosphatidylinositol-4-phosphate (PI(4)P; *Banerjee and Kane, 2017*). Mammals possess four subunit a isoforms (a1, a2, a3, and a4) which steer the V-ATPase to different endomembranes or the plasma membrane (*Forgac, 2007*; *Marshansky et al., 2014*; *Futai et al., 2019*). In Arabidopsis VHA-a2 and VHA-a3 target the V-ATPase to the tonoplast (*Dettmer et al., 2006*) where the combined action of V-ATPase and V-PPase energizes secondary active transport and maintains the acidic environment required for the lytic function of plant vacuoles (*Sze et al., 1999*; *Kriegel et al., 2015*). In contrast, VHA-a1 targets the V-ATPase to the TGN/EE, a highly dynamic organelle that receives and sorts proteins from the endocytic, recycling and secretory pathways (*Dettmer et al., 2006*; *Viotti et al., 2010*). The combined action of the V-ATPase and proton-coupled antiporters including the NHX-type cation proton exchangers (*Bassil et al., 2011a* and *Bassil et al., 2011b*; *Dragwidge et al., 2019*) and the ClC (Chloride channel) $Cl^-/NO^-$ proton antiporters (*von der Fecht-Bartenbach et al., 2007*) is responsible for generating and maintaining the acidic pH of the TGN/EE (*Luo et al., 2015*). Genetic and pharmacological inhibition of the V-ATPase interferes with endocytic and secretory trafficking (*Dettmer et al., 2006*; *Viotti et al., 2010*; *Luo et al., 2015*) and causes defects in both cell division and cell expansion (*Dettmer et al., 2006*; *Brüx et al., 2008*). Despite the TGN/EE being the central hub for protein sorting, it is still not clear how the identity of this compartment is specified and how resident proteins required for functionality are maintained while at the same time very similar proteins are sorted and leave the TGN/EE as cargo.

We have shown previously that after assembly involving dedicated ER-chaperones (*Neubert et al., 2008*), VHA-a1 and VHA-a3 containing Vo-subcomplexes leave the ER via different trafficking routes. Whereas VHA-a1 containing V-ATPases are exported in a coat protein complex II (COPII)-dependent manner (*Viotti et al., 2013*), VHA-a3 containing complexes have been shown to be delivered in a Golgi-independent manner to the tonoplast (*Viotti et al., 2013*). It remains to be addressed how V-ATPases containing the different VHA-a isoforms are sorted in the ER and how they are targeted to their final destinations in the cell. Here, we focus on VHA-a1 to understand how the V-ATPase is targeted to and retained in the TGN/EE. Targeting of TGN-resident proteins has been shown to be mediated by groups of acidic amino acids (acidic clusters), di-leucine and tyrosine-based motifs in mammals (*Bos et al., 1993*; *Schäfer et al., 1995*; *Alconada et al., 1996*; *Xiang et al., 2000*). In yeast, retrieval to the late Golgi (equivalent to the TGN) is dependent on the concerted action of aromatic-based amino acid (aa) motifs in the cytoplasmic tails of proteins and slow anterograde transport to the late endosome (*Wilcox et al., 1992*; *Nothwehr et al., 1993*; *Cooper and Stevens, 1996*; *Cereghino et al., 1995*; *Bryant and Stevens, 1997*). We have previously shown that the TGN/EE targeting information is contained within the first 228 aa of VHA-a1 (*Dettmer et al., 2006*). Here, by using chimeric proteins, 3D homology modeling, site-directed mutagenesis and live cell imaging, we identified a region that is required for ER-export as well as TGN/EE-retention. Although most plant genomes encode multiple VHA-a isoforms, the VHA-a1 targeting domain (a1-TD) is conserved only among seed plants. The liverwort *Marchantia polymorpha* with a single gene encoding subunit a (MpVHA-a) that is mostly present at the tonoplast in both Marchantia and Arabidopsis might represent the ancestral state in which acidification of the TGN/EE was achieved by transitory V-ATPase complexes. The identification of homozygous mutants lacking VHA-a1 indicates that the VHA-a isoforms, despite their differential localization, act redundantly. Based on the results presented here, we propose a model in which differential targeting of the VHA-a isoforms is based on competition for entry into COPII vesicles.

## Results

### Identification of a region in the VHA-a1 N-terminus (a1-TD) that is necessary and sufficient for TGN/EE localization

The tri-peptide motif that mediates Golgi-localization of Stv1p in yeast (*Finnigan et al., 2012*) is not conserved in VHA-a1 (*Figure 1—figure supplement 1*) implying that TGN/EE localization evolved independently. We thus used chimeric proteins consisting of increasing lengths of the cytosolic N-terminal domain of VHA-a1 (a1NT 37, 85, 131, 179 and 228 aa) fused to decreasing lengths of the C-terminal domain of VHA-a3, to further narrow down the region required for TGN/EE targeting.

Constructs encoding the chimeric proteins fused to GFP were expressed in Arabidopsis under control of the *UBIQUITIN10* promoter (*UBQ10*; *Grefen et al., 2010*). Whereas a1NT37a3-GFP, a1NT85a3-GFP (*Figure 1—figure supplement 2*) and a1NT131a3-GFP (*Figure 1A*) all localized to the tonoplast, a1NT179a3-GFP and a1NT228a3-GFP were detectable at the tonoplast but also in a punctate pattern reminiscent of the TGN/EE (*Figure 1A*). TGN/EE localization was confirmed by colocalization with the endocytic tracer FM4-64 (*Dettmer et al., 2006*) and treatment with the fungal drug Brefeldin A (BFA) that causes the aggregation of the TGN/EE to form BFA compartments (*Nebenführ et al., 2002*; *Dettmer et al., 2006*). After 3 hr of BFA treatment, the core of BFA compartments was labeled by a1NT179a3-GFP and a1NT228a3-GFP (*Figure 1B*). From these observations, we concluded that the region necessary for TGN/EE localization of VHA-a1 (a1-targeting domain; a1-TD) is located between residues L132 and E179.

To visualize the a1-TD in the three-dimensional (3D) structure, we used I-TASSER (*Roy et al., 2010*) to generate 3D models of the VHA-a1 and VHA-a3 N-termini with the atomic models of the yeast isoforms Stv1p and Vph1p (*Vasanthakumar et al., 2019*) serving as templates, respectively (*Figure 1C*, *Supplementary file 1* and *Figure 1—figure supplement 3*). Superimposition of the obtained models for the VHA-a1 and VHA-a3 N-termini revealed that the a1-TD is one of the regions predicted to be different (*Figure 1C*). To visualize the orientation of the VHA-a1 N-terminus within the V-ATPase complex, we aligned our model for the VHA-a1 N-terminus model with the structure of the yeast V-ATPase (PDB6O7V; *Vasanthakumar et al., 2019*) and found that the a1-TD would be accessible for recognition in the fully assembled V-ATPase complex (*Figure 1D*).

To test if the VHA-a1-TD is sufficient to target VHA-a3 to the TGN/EE, we introduced the 34 aa region between K140 and S174 into the N-terminus of VHA-a3. The resulting GFP fusion construct (*UBQ10:VHA-a3-a1-TD-GFP*) was transformed into wildtype plants and VHA-a3-a1-TD-GFP was detectable at the tonoplast and at the TGN/EE where it colocalized with VHA-a1-mRFP (*Figure 1E*). It remains to be determined why TGN/EE localization of VHA-a3-a1-TD is only partial, however it demonstrates that the a1-TD is sufficient to partially re-route VHA-a3 to the TGN/EE.

## Mutation of the a1-TD in VHA-a1 leads to mislocalization to the tonoplast

Phylogenetic analysis of VHA-a sequences from mono- and dicot species showed that VHA-a1 and VHA-a3 represent two clearly separated clades (*Figure 2—figure supplement 1*). The a1-TD includes a number of amino acids conserved among all of the sequences in the VHA-a1 clade (*Figure 2A*) but not in the VHA-a3 clade that could either be part of a di-acidic ER-export motif or an acidic cluster involved in TGN/EE-retention. Next, we used site-directed mutagenesis to identify which amino acids are required for correct localization of VHA-a1. Conserved amino acids in the a1-TD were exchanged individually or in combinations of two or three to the corresponding amino acid of VHA-a3. To assess the effect that each mutation had on the localization of VHA-a1-GFP, we performed colocalization experiments with VHA-a1-mRFP. Unmutated VHA-a1-GFP coexpressed with VHA-a1-mRFP was used as a control. Nine to ten images were subjected to quantitative image analysis. The overall degree of colocaliaztion was measured using the standard Pearson's correlation coefficient (*Manders et al., 1992*). In addition, to determine the fraction of VHA-a1-GFP in the VHA-a1-mRFP location, the Manders' Colocalization Coefficients (*Manders et al., 1993*) were also calculated for each individual image.

Mutations with high Pearson's correlation coefficients predominately localized to the TGN/EE (E161S and F134Y; *Figure 2C,D and K*). Three mutations led to a dual TGN/EE and tonoplast localization (L159T + E161S, L159T, E155 + E156 + I157 deletion (ΔEEI) and *Figure 2E,F,G and H*) and had intermediate Pearson's and Mander's coefficients ($M_2$) (*Figure 2K*). Mutation E156Q + L159T + E161S (ELE) also led to a dual TGN/EE and tonoplast localization but displayed fewer TGN/EEs therefore had a lower Mander's coefficient compared to the other mutations that had a dual TGN/EE and tonoplast localization. Two mutations led to VHA-a1 being localized to an unknown compartment (E156Q and E156Q + L159T; *Figure 2I and J*) and had the lowest Pearson's and Mander's coefficients (*Figure 2K*).

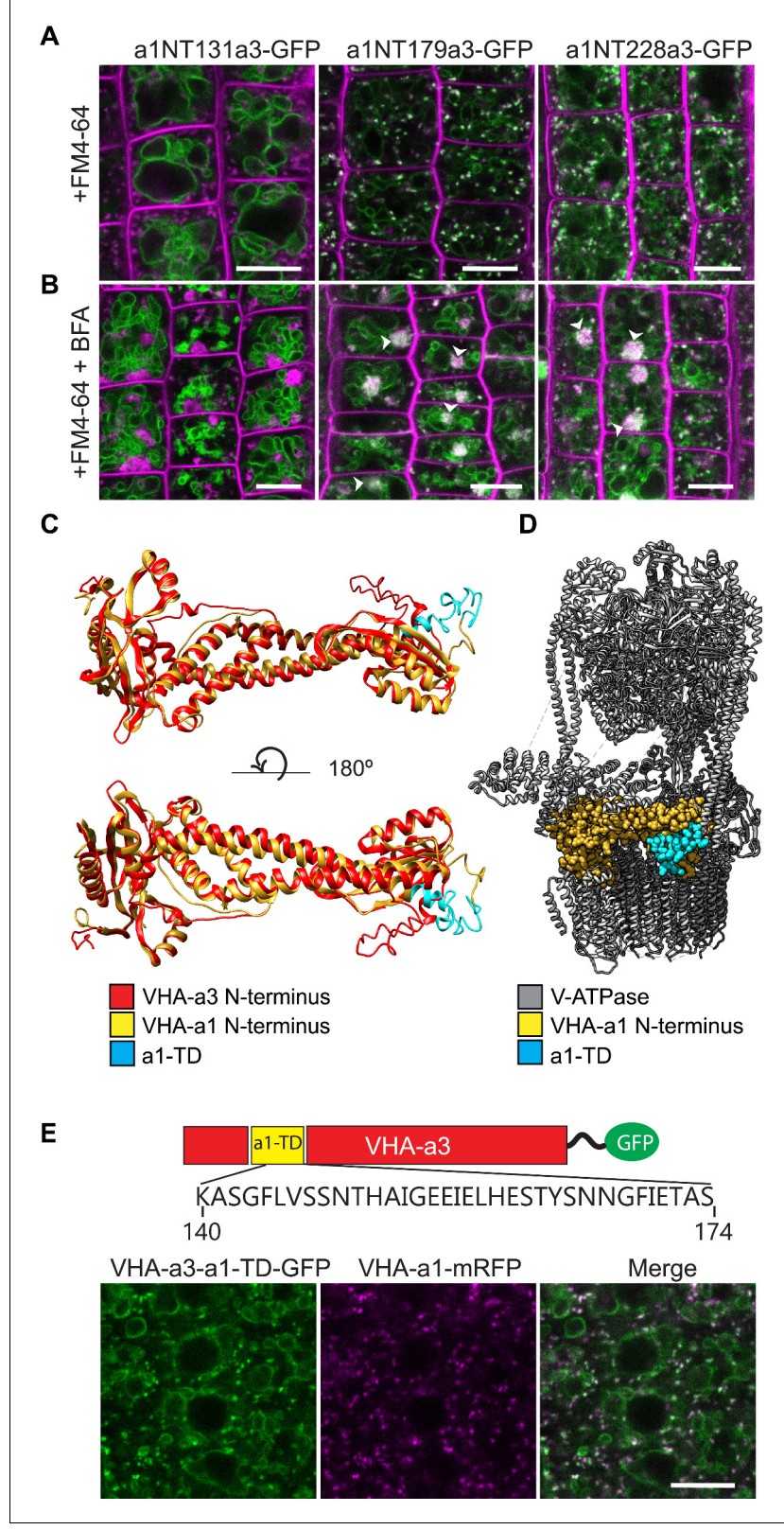

**Figure 1.** The targeting signal of VHA-a1 is located between L132 and E179 and the a1-TD is sufficient for targeting of VHA-a3 to the TGN/EE. Root tips of 6-day-old Arabidopsis seedlings were analyzed by confocal laser scanning microscopy (CLSM) (**A**) The chimeric constructs a1NT179a3-GFP and a1NT228a3-GFP show dual localization at the TGN/EE and tonoplast. (**B**) TGN/EE localization of a1NT179a3-GFP and a1NT228a3-GFP was

*Figure 1 continued on next page*

*Figure 1 continued*

confirmed by treatment of Arabidopsis root cells with 50 µM BFA for 3 hr followed by staining with FM4-64 for 20 min. Scale bars = 10 µm. Green and magenta pseudo colors indicate fluorescence from GFP and FM4-64 respectively. (C) One of the structural differences between the VHA-a1 NT and VHA-a3 NT corresponds to the location of the a1-TD (blue). (D) An alignment of the VHA-a1 NT model with a model of one of the rotational states of the yeast V-ATPase (PDB = 6O7V) revealed that the VHA-a1 targeting domain is exposed and accessible for recognition. (E) VHA-a3 with the a1-TD (VHA-a3-a1TD-GFP) partially colocalizes with VHA-a1-mRFP. Scale bar = 10 µm.

The online version of this article includes the following figure supplement(s) for figure 1:

**Figure supplement 1.** The tri-peptide motif that is responsible for the targeting of Stv1p is absent in VHA-a1.

**Figure supplement 2.** The targeting signal of VHA-a1 is not located in the first 85 amino acids.

**Figure supplement 3.** Three-dimensional models of the VHA-a1 and VHA-a3 N-termini.

## The a1-TD is required for COPII-mediated ER-export

As we have shown previously that VHA-a3 containing complexes are delivered in a Golgi-independent manner from the ER to the tonoplast (*Viotti et al., 2013*), we next asked if the tonoplast signal observed for the mutated VHA-a1 proteins was due to poor retention at the TGN/EE or partial entry into the provacuolar route caused by reduced ER-exit.

To address if the a1-TD contains an ER-export motif, we made use of a dominant negative mutation of Sar1 (Sar1-GTP) that has been shown to block COPII-mediated ER-export (*daSilva et al., 2004*). We expressed AtSar1b-GTP-CFP under a dexamethasone (DEX) inducible promoter (*pUBQ10 >GR > AtSar1b-GTP-CFP*; *Moore et al., 1998*) and used electron microscopy (EM) to confirm that COPII-mediated ER-export was indeed blocked. Upon induction of AtSar1b-GTP-CFP, we observed large clusters of vesicles in the periphery of Golgi stack remnants and ER cisternae appeared swollen (*Figure 3—figure supplement 1*). Furthermore, in Arabidopsis transgenic lines expressing the Golgi-marker Sialyl transferase (ST; *Boevink et al., 1998*) or the brassinosteroid receptor (BRI1; *Geldner et al., 2007*) as GFP fusion proteins, induction of AtSar1b-GTP-CFP caused their ER-retention and colocalization with AtSar1b-GTP-CFP in bright and often large punctae that might represent the clusters of uncoated COPII vesicles as observed by EM (*Figure 3—figure supplement 1*).

Similarly, the punctate pattern of VHA-a1-GFP disappeared after induction of AtSar1b-GTP-CFP and was replaced by a characteristic ER pattern with dense punctae that colocalized with AtSar1b-GTP-CFP (*Figure 3A*). In contrast, VHA-a3-GFP was not retained in the ER after the induction of AtSar1b-GTP-CFP (*Figure 3B*). To exclude that these differences in ER retention between VHA-a1-GFP and VHA-a3-GFP were due to differences in induction strength of AtSar1b-GTP-CFP, we also blocked ER exit in a line co-expressing VHA-a1-GFP and VHA-a3-mRFP. In cells in which VHA-a1-GFP was retained in the ER, VHA-a3-mRFP was not affected (*Figure 3C*). Conversely, VHA-a3-a1-TD-GFP is partially retained in the ER upon expression of AtSar1b-GTP-CFP (*Figure 3D*). Taken together, our data confirms that the ER-export of VHA-a1 is COPII-dependent and reveals that the a1-TD contains an ER-export signal.

When AtSar1b-GTP was used to block ER exit, we observed a significant increase in the GFP fluorescence intensity at the tonoplast for all the mutated VHA-a1 proteins (*Figure 4A and B*) indicating that blocking ER-export causes an increase in the trafficking to the tonoplast via a Golgi-independent route.

Based on our observation that VHA-a3 carrying a mutation (VHA-a3-R729N-GFP) that renders the complex inactive is retained in the ER when expressed in wildtype but found at the tonoplast in the *vha-a2 vha-a3* mutant (*Figure 5—figure supplement 1*; *Krebs et al., 2010*), we next tested if tonoplast localization of proteins carrying mutations in the a1-TD increases in the absence of competing complexes containing VHA-a2 and VHA-a3. Indeed, when the mutated VHA-a1 proteins were expressed in the *vha-a2 vha-a3* mutant, the ratio of TGN/EE-to-tonoplast fluorescence intensity increased as compared to the wildtype background (*Figure 5A and B*). Similarly, VHA-a1 mutants that showed only TGN localization (E161S) or a punctate pattern that is different from VHA-a1-RFP (E156Q and E156Q+ L159T) in the wildtype background displayed a tonoplast signal in the *vha-a2 vha-a3* double mutant (*Figure 5A and B*). These results strongly support the idea that all mutations

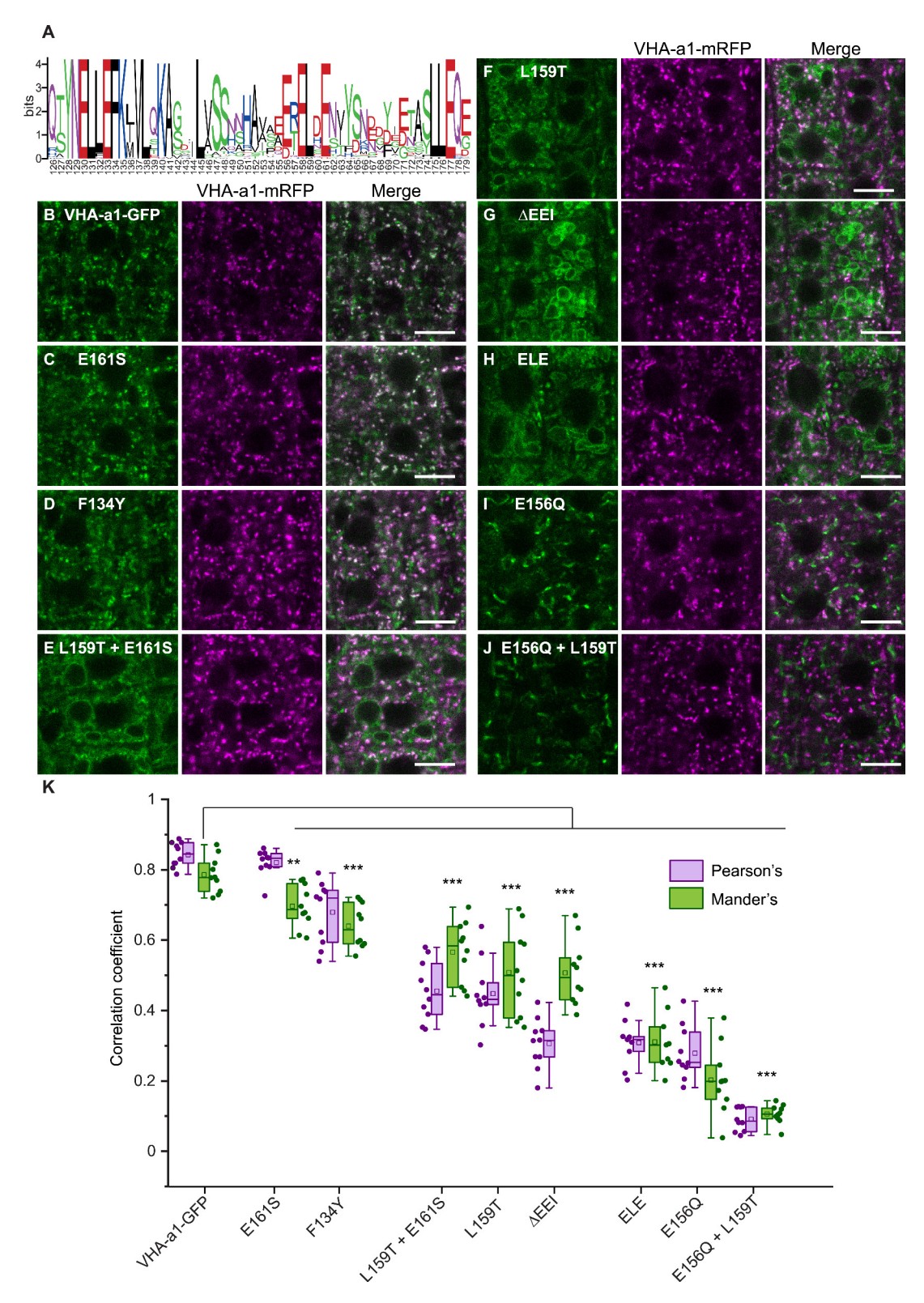

**Figure 2.** Site-directed mutagenesis reveals the importance of conserved amino acids in the targeting of VHA-a1 to the TGN/EE. (**A**) The VHA-a1-clade consensus sequence for the a1-TD region was made on the weblogo platform (*Crooks et al., 2004*). Sequence numbers are based on Arabidopsis VHA-a1. Conserved amino acids were mutated and their effect on the localization of GFP tagged VHA-a1 was analyzed in the VHA-a1-mRFP background. Root tips of 6-day-old Arabidopsis seedlings were analyzed by CLSM. (**B**) VHA-a1-GFP was co-expressed with VHA-a1-mRFP as a control.
*Figure 2 continued on next page*

*Figure 2 continued*

The mutations in VHA-a1 produced three classes of punctate patterns that could be classified as TGN/EE only (**C and D**), TGN/EE and tonoplast (**E, F, G and H**) and different from VHA-a1-mRFP (**I and J**) Scale bars = 10 μm. Green and magenta pseudo colors indicate fluorescence from GFP and VHA-a1-mRFP respectively. (**K**) Mutated VHA-a1-GFP proteins colocalization with VHA-a1-mRFP as shown by Pearson's and Mander's coefficients. The Mander's coefficient indicates the fraction of mutated VHA-a-GFP in the VHA-a1-mRFP location. Box plot center lines, medians; center boxes, means with n = 9–10 ratios calculated from 9 to 10 images; box limits, 25th and 75th percentiles; whiskers extend to ±1.5 interquartile range. Asterisks indicate significant differences of the mean Mander's correlation coefficients between mutated VHA-a1-GFP and VHA-a1-GFP (Two-sample $t$-Test, $p < 0.05$) (*$p < 0.05$, **$p < 0.01$ and ***$p < 0.001$).

The online version of this article includes the following source data and figure supplement(s) for figure 2:

**Source data 1.** Source data for *Figure 2K*.
**Figure supplement 1.** The VHA-a1-TD is conserved in Angiosperms.

affect ER export and that the observed tonoplast signal is not a result of poor retention at the TGN/EE but due to a Golgi-independent pathway to the tonoplast.

## The a1-TD is conserved in the plant kingdom and originates with the gymnosperms

To trace the evolutionary origin of the a1-TD in the plant kingdom, VHA-a sequences from selected species were subjected to phylogenetic analysis. Our analysis revealed that all VHA-a isoforms from seed plants including the gymnosperm *Pinus taeda* cluster into two distinct clades containing VHA-a1 and VHA-a3 respectively (*Figure 6—figure supplement 1* and *Figure 6A*). Interestingly, the genomes of most basal plants encode multiple isoforms of VHA-a, however they do not fall into either of these two clades implying that duplication of VHA-a occurred independently. Closer analysis of the a1-TD sequence revealed that the hallmarks of the a1-TD (acidic residues flanking a critical leucine residue) are conserved throughout the angiosperms but are absent in bryophytes, lycophytes, ferns and hornworts. Interestingly, the gymnosperm sequence from *P. taeda* contains the a1-TD sequence with the exception of one flanking glutamic residue (*Figure 6B*). To test if the a1-TD is functionally conserved we generated chimeric proteins consisting of the N-terminal domain of VHA-a from *Pinus taeda* and *Amborella trichopoda* as representatives of the seed plants and *Selaginella moellendorffii* as a non-seed plant fused to the C-terminal domain of VHA-a1 (*Figure 6C*). Whereas the Pine and Amborella chimeric proteins colocalized with VHA-a1 at the TGN/EE, the Selaginella chimeric protein localized to the tonoplast. These results suggest that the a1-TD domain is conserved within the spermatophytes but not in other green plants.

## Mislocalized VHA-a1 can replace the tonoplast V-ATPase

Mislocalization of VHA-a1 at the tonoplast provides the opportunity to address if the isoforms belonging to the VHA-a1 and VHA-a3 clades are functionally divergent. We used the ability to complement the dwarfed phenotype of the *vha-a2 vha-a3* double mutant (*Krebs et al., 2010*) in standard long day conditions (*Figure 7—figure supplement 1*) as well as in short day conditions (*Figure 7A*) as a proxy. We confirmed that the proteins were expressed (*Figure 7—figure supplement 2*) and observed that complementation of the growth phenotype correlated with the intensity of the tonoplast signal with VHA-a1E161S-GFP conferring the lowest and VHA-a1ΔEEI-GFP the highest degree of rescue (*Figure 7A and B*). The ability of V-ATPases with incorporated mutated VHA-a1-GFP subunits to complement the *vha-a2 vha-a3* double mutant rosette phenotype is an indication that these complexes localize to the tonoplast and are functional in all plant tissues and not only in root cells. Three VHA-a1-GFP mutation lines showing high (ΔEEI), intermediate (E156Q + L159T + E161S) and low degrees (L159T + E161S) of rescue were selected for further analysis. Cell sap pH measurements were performed as a proxy of vacuolar pH. Whereas all VHA-a1-GFP mutation lines had more acidic vacuoles than the *vha-a2 vha-a3* mutant none of them reached wildtype vacuolar pH (*Figure 7C*). Taken together, our data indicates that although VHA-a1 containing complexes are functional when mistargeted to the tonoplast their enzymatic properties are different and they cannot fully replace the vacuolar isoforms.

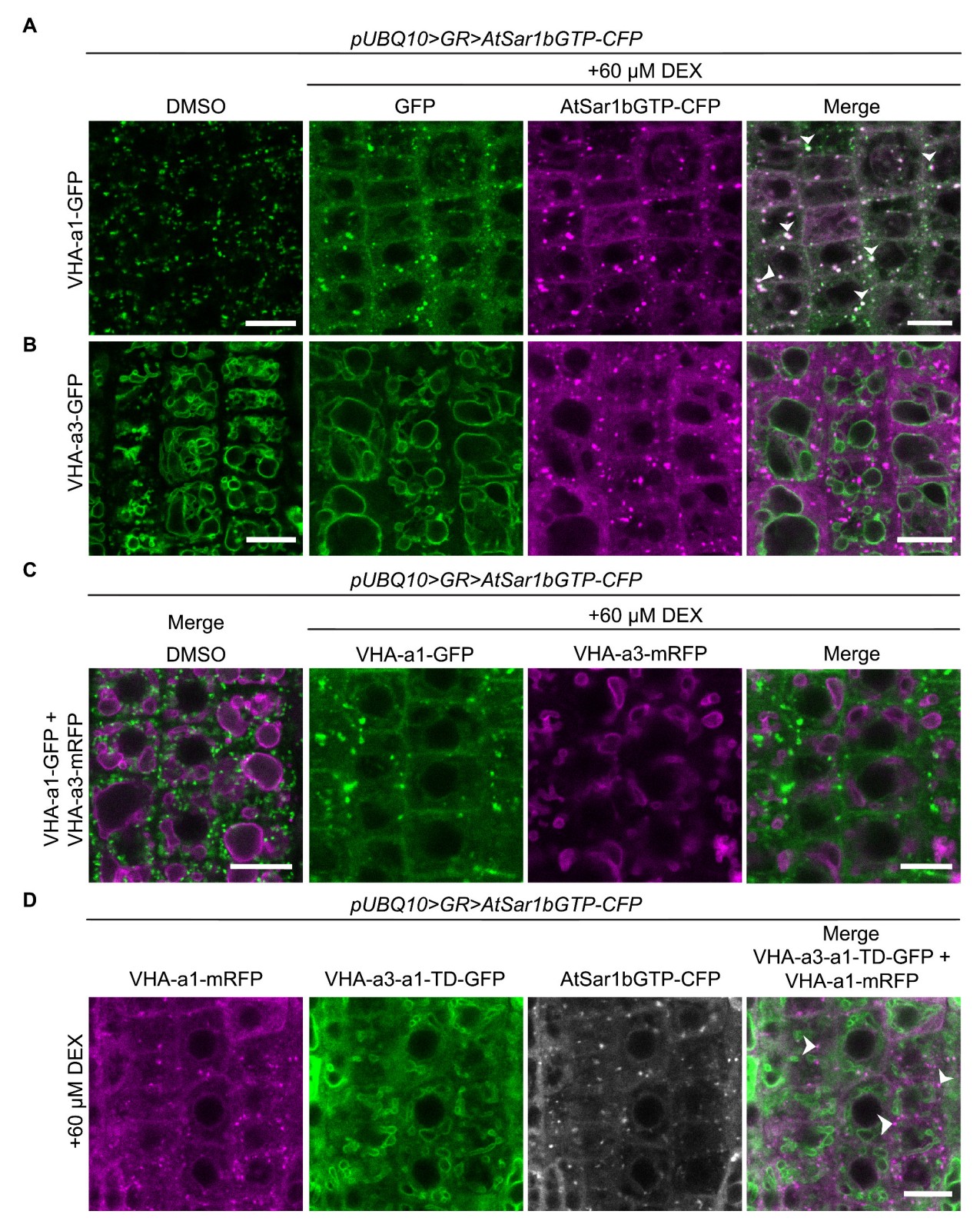

**Figure 3.** VHA-a1 is retained at the ER after AtSar1b-GTP-CFP expression. After 6 hr of induction with 60 µM DEX, AtSar1b-GTP-CFP is expressed in 6-day-old Arabidopsis root tip cells. (**A**) VHA-a1-GFP is retained in the ER and also agglomerates to produce bright punctae which colocalize with AtSar1b-GTP-CFP (white arrows). For the DMSO control, only the GFP channel is shown. (**B**) VHA-a3-GFP does not accumulate when exit from the ER via COPII vesicles is blocked by expression of AtSar1b-GTP-CFP. For the DMSO control, only the GFP channel is shown. (**C**) When VHA-a1 and VHA-a3

*Figure 3 continued on next page*

Figure 3 continued

are co-expressed in the same cell only VHA-a1 accumulates in the ER upon induction of AtSar1b-GTP-CFP. (D) VHA-a3-a1-TD-GFP is partially retained in the ER after induction of AtSar1b-GTP-CFP. Arrows point to VHA-a1-mRFP aggregates. Scale bars = 10 μm.

The online version of this article includes the following figure supplement(s) for figure 3:

**Figure supplement 1.** AtSar1b-GTP-CFP expression blocks the ER exit of secretory pathway proteins.

## The Marchantia V-ATPase is dual localized at the TGN/EE and tonoplast and is functional at the tonoplast in Arabidopsis

Most plants outside of the spermatophytes possess multiple VHA-a isoforms with the exception of the liverwort *Marchantia polymorpha* and the fern *Salvinia cucullata* (*Supplementary file 1*). Focusing on Marchantia, we addressed if the TGN/EE or the tonoplast V-ATPase represent the ancestral state or if dual localization of the V-ATPase can also be achieved in the absence of differentially localized isoforms. To examine the subcellular localization of the single, *M. polymorpha* VHA-a protein (MpVHA-a), mVenus-tagged MpVHA-a (MpVHA-a-mVenus) was co-expressed with the TGN marker, mRFP-MpSYP6A (*Kanazawa et al., 2016*) in *M. polymorpha* thallus cells. *MpVHA-a-mVenus* driven by the *CaMV35S* or *MpEF1α* promoter predominantly localized to the vacuolar membrane, but some additional punctate structures were detectable (*Figure 8—figure supplement 1*).

Next, we expressed a MpVHA-a-mVenus fusion construct (*UBQ10:MpVHA-a-mVenus*) in the Arabidopsis wildtype background. Whereas MpVHA-a-mVenus localization at the tonoplast of root cells was clearly visible (*Figure 8—figure supplement 2*), a clear punctate pattern was not observed. However, after BFA treatment, the core of BFA compartments was labeled with MpVHA-a-mVenus (*Figure 8—figure supplement 2*) indicating that MpVHA-a is also present at the TGN/EE. To test for functionality, MpVHA-a-mVenus was expressed in the *vha-a2 vha-a3* mutant background. MpVHA-a-mVenus localized to the TGN/EE and tonoplast and was found in the core of FM4-64 labeled BFA compartments (*Figure 8A*). A growth assay conducted in short day conditions (SD; 22 ° C and 10 hr light) revealed that V-ATPases that incorporate MpVHA-a can complement the *vha-a2 vha-a3* double mutant to wildtype levels in terms of rosette size and cell sap pH (*Figure 8B and C*). We also confirmed that the MpVHA-a-mVenus proteins are expressed (*Figure 8D*).

## VHA-a1 has a unique and essential function during pollen development which cannot be fulfilled by VHA-a2, VHA-a3 or MpVHA-a

Next, we wanted to test if MpVHA-a can also replace VHA-a1 at the TGN/EE. Null alleles of single copy-encoded VHA-subunits cause male gametophyte lethality (*Dettmer et al., 2005*) and it has thus not been possible to identify homozygous T-DNA mutants for VHA-a1. To avoid T-DNA related silencing problems in complementation experiments with heterozygous *vha-a1/+* mutants, we used CRISPR/Cas9 under control of an egg cell-specific promoter to generate mutant alleles (*Wang et al., 2015*). Different regions of *VHA-a1* were targeted using four different guide RNAs (gRNAs 1–4, *Figure 9A*). Wildtype plants as well as *VHA-a1-GFP* expressing transgenic plants were transformed and T1 plants were analyzed by sequencing of PCR products spanning the CRISPR site. Four alleles were selected for further studies: *vha-a1-1* containing a 260 bp deletion which eliminates the start codon and *vha-a1-2,−3* and *−4* that each contain single base pair insertions leading to frameshifts and early stop codons (*Figure 9A*).

Surprisingly, we did not only identify heterozygous *vha-a1/+* but also homozygous and bi-allelic *vha-a1* individuals (*Supplementary file 1*). The latter two were indistinguishable from wildtype during vegetative growth (*Figure 9B*), but failed to produce seeds due to a defect in pollen development (*Figure 9C*). To confirm that the defect in pollen development is indeed caused by a lack of VHA-a1, we analyzed if it is rescued by *VHA-a1-GFP*. Although *VHA-a1:VHA-a1-GFP* (*Dettmer et al., 2006*) is also targeted by the chosen gRNAs and *UBQ10:VHA-a1-GFP* is targeted by gRNAs 2,3 and 4, the use of appropriate primer combinations allowed us to distinguish between mutations in the endogenous locus and the transgene (*Figure 9—figure supplement 1*). Plants that carried mutations corresponding to *vha-a1-2* in the transgene did not express VHA-a1-GFP confirming that it is indeed a null allele (*Figure 9—figure supplement 1*). Importantly, all plants with no wildtype allele of either *VHA-a1* or *VHA-a1-GFP* were defective in pollen development,

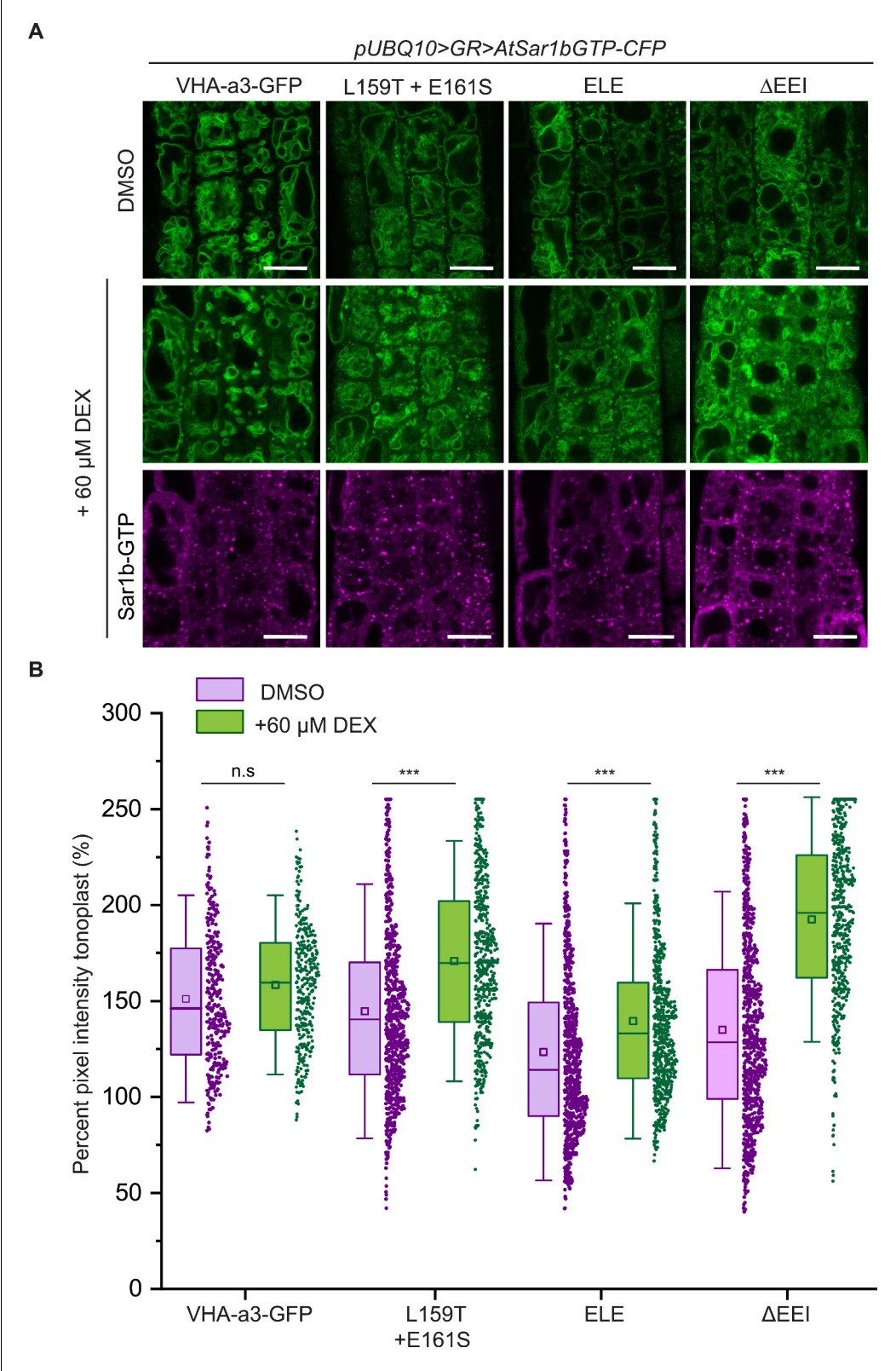

**Figure 4.** The mutations in the a1-TD affect an ER-exit motif. The localization and fluorescence intensity of mutated VHA-a1-GFP proteins was analyzed in the presence (+DEX) and absence (DMSO) of AtSar1b-GTP-CFP. Root tips of 6-day-old Arabidopsis seedlings were used for the analysis. (A) The mutated VHA-a1 proteins still localized at the tonoplast when exit from the ER via COPII vesicles was blocked by expression of AtSar1b-GTP-CFP. Green and magenta pseudo colors indicate fluorescence from GFP and CFP respectively Scale bars = 15 µm. (B) The tonoplast fluorescence intensity

*Figure 4 continued on next page*

*Figure 4 continued*

was measured in the presence and absence of DEX. There is a significant increase in the GFP fluorescence intensity at the tonoplast of mutated VHA-a1 proteins when exit from the ER via COPII vesicles is blocked by expression of AtSar1b-GTP-CFP. Box plot center lines, medians; center boxes, means with n ≥ 319 measurements; box limits, 25th and 75th percentiles; whiskers extend to ±1.5 SD. Asterisks indicate significant differences between the uninduced and induced conditions (Mann-Whitney test, $p<0.001$). (n.s: not significant,$*p<0.05$, $**p<0.01$ and $***p<0.001$).

The online version of this article includes the following source data for figure 4:

**Source data 1.** Source data for *Figure 4B*.

whereas *vha-a1* mutants expressing *VHA-a1-GFP* under the control of *UBQ10* promoter showed normal pollen development (*Figure 9C*).

We next established stable lines expressing *UBQ10:VHA-a1-GFP in the vha-a1-1* background and performed reciprocal crosses to determine if transmission via the female gametophyte is also affected. As expected, *vha-a1-1* was only transmitted via the male gametophyte in the presence of *VHA-a1-GFP*. In contrast, transmission of *vha-a1-1* via the female gametophyte did not require the presence of the transgene indicating that development of the female gametophyte is not affected (*Supplementary file 1*).

As homozygous *vha-a1* mutants cannot be transformed, rescue experiments would require crosses with a pollen donor that carries a wildtype allele of *VHA-a1* so that complementation could only be determined in the F2. For crosses we thus used *vha–a1* mutants that still contained the CRISPR T-DNA (Cas9+) so that the incoming wildtype allele would be mutated and homozygous *vha-a1* mutants could be obtained in the resulting F1. Using this strategy, we identified bi–allelic and homozygous *vha-a1* mutants expressing *UBQ10:MpVHA-a-mVenus* and found that it was localized both at the TGN/EE and the tonoplast (*Figure 9—figure supplement 2*). However, MpVHA-a-mVenus did not rescue the pollen phenotype of *vha-a1* (*Figure 9—figure supplement 2*) indicating that the TGN/EE-localized V-ATPase of seed plants has acquired a unique function during pollen development.

## During vegetative growth V-ATPases containing VHA-a2 and VHA-a3 compensate for the lack of VHA-a1

Given the importance of acidification of the TGN/EE for endomembrane trafficking and the fact that we have previously reported that RNA-mediated knock-down of VHA-a1 causes reduced cell expansion (*Brüx et al., 2008*), the observation that the *vha-a1* mutant displays no different phenotypes from wildtype during vegetative growth is unexpected. However, *vha-a1* is hypersensitive to the V-ATPase inhibitor Concanamycin-A (ConcA). Analysis of the progeny of *vha-a1 VHA-a1-GFP (Cas9+)* mutants revealed that *vha-a1 vha-a1-GFP* etiolated seedlings displayed shorter hypocotyls and had deformed roots in contrast to *vha-a1* mutants expressing *VHA-a1-GFP*. This result indicates that a target of ConcA is present at the TGN/EE in *vha-a1* albeit in small amounts or with a higher sensitivity to ConcA. (*Figure 10A* and *Figure 10—figure supplement 1*). To test if VHA-a2 or VHA-a3-containing V-ATPases might compensate for the lack of VHA-a1, *vha-a1-1 (Cas9+)* was crossed with the *vha-a2 vha-a3* double mutant. *vha-a1 vha-a2/+ vha-a3/+* mutants obtained from this cross were significantly smaller than *vha-a2/+ vha-a3/+* individuals indicating that tonoplast V-ATPases indeed compensate for the lack of VHA-a1 during vegetative growth (*Figure 10B* and *Figure 10—figure supplement 2*).

Subsequently, *vha-a1 (Cas9+)* was crossed with *vha-a2* and *vha-a3* single mutants to determine if both are able to compensate for the lack of VHA-a1. *vha-a1 vha-a2/+ and vha-a1 vha-a3/+* plants both had reduced rosette sizes, however the latter showed a stronger reduction which is in accordance with VHA-a3 being expressed at higher levels than VHA-a2 (*Figure 10—figure supplement 2*). To analyze mutants with even lower numbers of *VHA-a* wildtype alleles *vha-a1 (Cas9+) vha-a2/+ vha-a3/+* was crossed with the *vha-a2 vha-a3* double mutant. Consistent with the results of the single mutant crosses, *vha-a1/+ vha-a2/+ vha-a3* was smaller than *vha-a1/+ vha-a2 vha-a3/+*. The smallest plants obtained were *vha-a1 vha-a2 vha-a3/+* (*Figure 10C*). Neither did we find *vha-a1 vha-a2/+ vha-a3* nor the homozygous triple mutant *vha-a1 vha-a2 vha-a3*, suggesting that they are not viable.

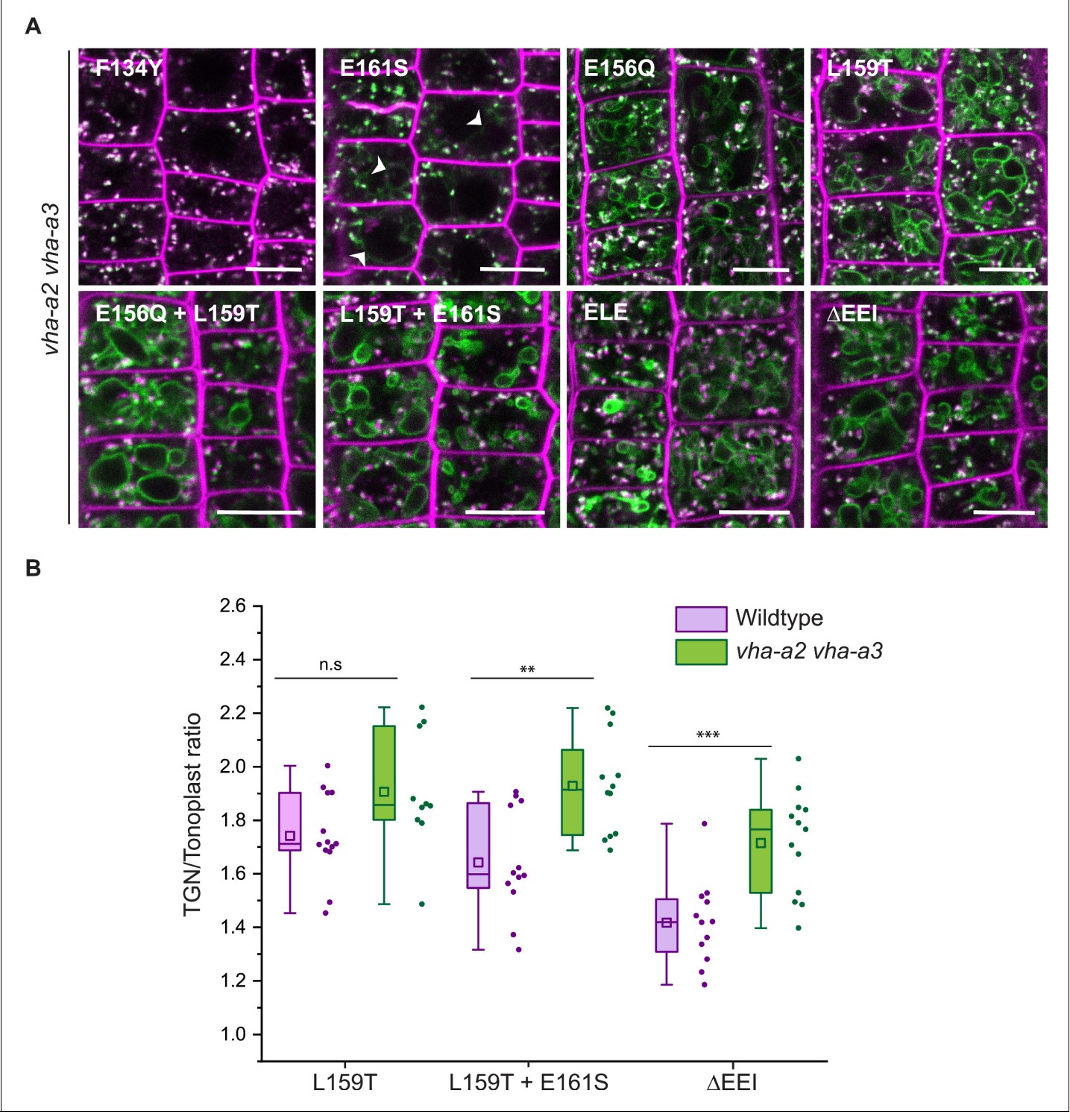

**Figure 5.** Tonoplast localization of mutated VHA-a1 increases in the absence of VHA-a2 and VHA-a3. (**A**) The localization of mutated VHA-a1 proteins tagged to GFP were analyzed in the *vha-a2 vha-a3* double mutant background after 20 min staining with FM4-64. Root tips of 6-day-old Arabidopsis seedlings were used for the analysis. Scale bars = 10 µm. Green and magenta pseudo colors indicate fluorescence from GFP and FM4-64 respectively. (**B**) TGN/EE-to-tonoplast fluorescence intensity ratios were calculated for selected mutations in the wildtype background and in the *vha-a2 vha-a3* double mutant background. The ratio of TGN/EE-to-tonoplast fluorescence intensity of the mutated VHA-a1 proteins (L159T + E161S and ΔEEI) is significantly higher in the *vha-a2 vha-a3* double mutant as compared to the wildtype background. Box plot center lines, medians; center boxes, means with n ≥ 10 ratios calculated from individual images; box limits, 25th and 75th percentiles; whiskers extend to ±1.5 interquartile range. Asterisks indicate

*Figure 5 continued on next page*

*Figure 5 continued*
significant differences between the wildtype and *vha-a2 vha-a3* double mutant mean ratios (Two-sample *t*-Test, p<0.05). (n.s: not significant, *p<0.05, **p<0.01 and ***p<0.001).
The online version of this article includes the following source data and figure supplement(s) for figure 5:

**Source data 1.** Source data for *Figure 5B*.
**Figure supplement 1.** A competition exists to enter provacuoles in wild type *Arabidopsis* root tip cells.

## Discussion

### The information needed to deliver and keep the V-ATPase at the TGN/EE is contained in the a1-TD

The compartments of the eukaryotic endomembrane system are acidified to varying degrees by the activity of the vacuolar $H^+$-ATPase (V-ATPase). Acidification enables specific biochemical reactions as well as secondary active transport and can thus be considered a central component of compartmentation. Throughout eukaryotes, isoforms of the membrane-integral $V_O$-subunit a are used to achieve differential targeting of the V-ATPase and although we know very little about the underlying mechanisms, it is clear that differential targeting has arisen independently in different groups. Whereas the targeting information is contained in the N-terminus of the Golgi-localized yeast isoform Stv1p (*Finnigan et al., 2012*), it was shown to be provided by the C-terminal part for some of the 17 isoforms of *Paramecium tetraurelia* that localize the V-ATPase to at least seven different compartments (*Wassmer et al., 2006*). Moreover, the targeting motif identified in Stv1p (*Finnigan et al., 2012*) is neither conserved in plant nor mammalian subunit a isoforms implying that differential targeting mediated by subunit a evolved independently in the different eukaryotic lineages. Given the essential role that the V-ATPase plays in endocytic and secretory trafficking (*Dettmer et al., 2006*; *Luo et al., 2015*), we set out to identify the mechanism underlying its highly specific localization at the TGN/EE (*Dettmer et al., 2006*). Based on our previous studies, in which we had shown that the targeting information is located within the 228 amino acids of VHA-a1 (*Dettmer et al., 2006*), we first narrowed down this region to a 30 aa region (a1-TD) which we show here to be both necessary and sufficient for TGN/EE localization. Structural modeling revealed that the a1-TD that contains several highly conserved acidic amino acids is exposed and accessible for interaction partners. Acidic clusters have been shown to be involved in TGN localization (*Schäfer et al., 1995*; *Alconada et al., 1996*; *Xiang et al., 2000*) and we thus used site-directed mutagenesis and found that mutation of conserved aa in the a1-TD indeed leads to partial mislocalization of VHA-a1 to the tonoplast as expected if retention at the TGN/EE would be affected. However, the acidic residues in the a1-TD could also represent di-acidic ER-export motifs and as we have shown previously that VHA-a3 is targeted to the tonoplast in a Golgi-independent manner (*Viotti et al., 2013*), we needed to differentiate between the a1-TD serving as an ER exit motif and/or a TGN-retention motif. We used inducible expression of the dominant negative GTPase; Sar1BH74L to block ER exit and observed that the mutated VHA-a1 proteins were not retained in the ER and importantly that the signal at the tonoplast was increased. Assuming that the a1-TD contains one or several overlapping diacidic-ER-exit motifs and no TGN-retention signal, it should re-route VHA-a3 into COPII- and Golgi-dependent trafficking to the tonoplast. However, a dual localization was observed for VHA-a3-a1-TD indicating either that the a1-TD contains a TGN-retention motif or that VHA-a3 has a yet unknown feature that actively sorts it into the provacuolar route.

### Which VHA-a isoform came first, TGN/EE or tonoplast and are they redundant?

Phylogenetic analysis revealed that the VHA-a isoforms of all angiosperms as well as the gymnosperm *P. taeda* fall into two distinct clades. The a1-TD is conserved within the VHA-a1 clade and exchange of the N-terminal domains within this clade resulted in TGN/EE localization whereas the N-terminal domain of Selaginella that lacks the a1-TD resulted in tonoplast localization indicating that the duplication leading to differential targeting occured in a common ancestor of all seed plants. With the exception of Marchantia and *Salvina culculatta*, all other plants outside of the spermatophytes possess multiple VHA-a isoforms, however the a1-TD is not conserved in these plants.

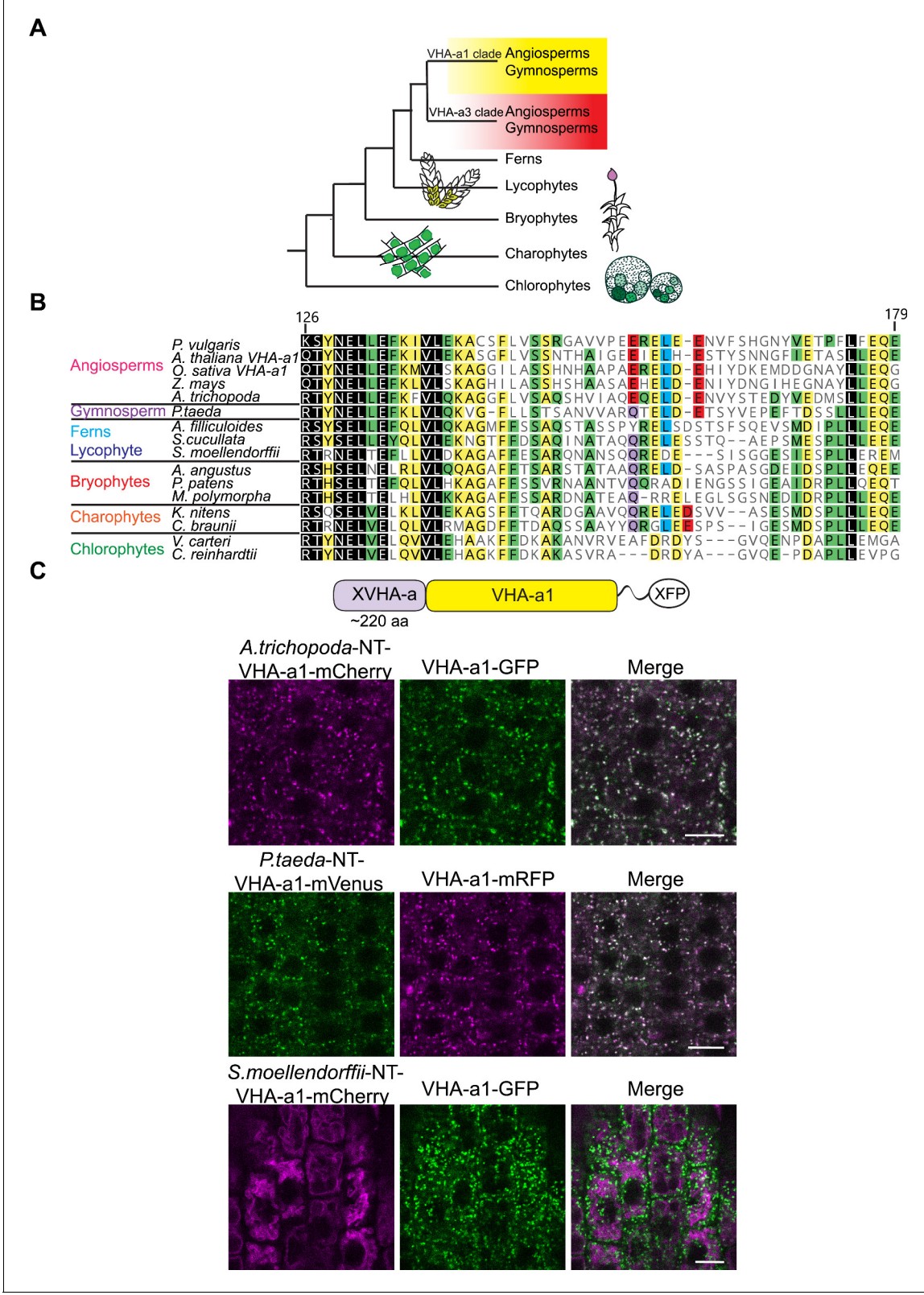

**Figure 6.** Spermatophyte VHA-a isoforms cluster into two distinct clades, the a1-TD is conserved in the VHA-a1 clade and originates in the gymnosperm sequences. (**A**) Phylogenetic analysis of the N-terminal sequences of VHA-a proteins was done. A graphical summary of the tree is depicted. VHA-a isoforms from seed plants including the gymnosperm *Pinus taeda* cluster into two distinct clades. (**B**) The a1-TD originates in the gymnosperms and it is absent from the chlorophytes, bryophytes, lycophytes and pteridophytes. The sequence of the gymnosperm, *P. taeda* and that
*Figure 6 continued on next page*

*Figure 6 continued*

of the charophytes contain all the amino acids thought to comprise the ER-exit signal with the exception of one flanking glutamic acid residue. The sequence numbers are in reference to the *A. thaliana* VHA-a1 sequence. (**C**) The a1-TD is functionally conserved in gymnosperms and angiosperms. Root tips of 6-day-old Arabidopsis seedlings were analyzed by CLSM. Pine and Amborella chimeric proteins colocalized with VHA-a1 at the TGN/EE. The Selaginella chimeric protein localized to the tonoplast only. Scale bars = 10 μm.

The online version of this article includes the following figure supplement(s) for figure 6:

**Figure supplement 1.** Phylogenetic analysis of the N-terminal sequences of VHA-a related proteins.

Evidence for differential targeting of VHA-a isoforms is missing but it seems reasonable to speculate that differential localization arose independently. Based on this, Marchantia might represent the ancestral ground state with a single VHA-isoform predominantly localized at the tonoplast. The trafficking machinery of Marchantia has distinct features and it remains to be determined if acidification of the TGN/EE plays a similar role as in Arabidopsis. If so, Marchantia would indeed represent a unique model for a single VHA-a isoform being sufficient for V-ATPase activity in two locations. A comparable situation has been proposed for the ancestral form of Stv1p and Vph1p that also served both Golgi- and vacuole acidification based on slow anterograde post-Golgi trafficking (*Finnigan et al., 2011*). In support of this notion, it has recently been shown that V-ATPase complexes containing Vph1p contribute more to Golgi-acidification while en route to the vacuole than the Golgi-localized complexes containing Stv1p (*Deschamps et al., 2020*). The identity of the VHA-a isoform does not only determine the subcellular localization, it can influence a number of biochemical properties including assembly, coupling efficiency, protein abundance and reversible dissociation (*Leng et al., 1998*; *Kawasaki-Nishi et al., 2001b*, *Kawasaki-Nishi et al., 2001a*).

Given the central role of the V-ATPase in many cellular functions it is important to not only understand the sorting of this molecular machine but also to address the functional diversification of plant VHA-a isoforms. The fact that VHA-a1 subunits with mutations in the a1-TD which are mislocalized to the tonoplast can complement the dwarf phenotype of the *vha-a2 vha-a3* mutant to varying degrees argues strongly that they have retained their basal proton-pumping activity and do not require the specific lipid-environment of the TGN/EE for functionality. However, VHA-a1 containing V-ATPases at the vacuole cannot acidify the vacuole to wildtype levels and it remains to be determined if this is caused eg. by a difference in coupling efficiency (*Kawasaki-Nishi et al., 2001b*) or pH-dependent feedback regulation (*Rienmüller et al., 2012*). In contrast, when the Marchantia VHA-a was expressed in *vha-a2 vha-a3*, the resulting transgenic lines were indistinguishable from wildtype. As null alleles of single copy-encoded VHA-subunits cause male gametophyte lethality (*Dettmer et al., 2005*), we assumed that we would have to transform heterozygous *vha-a1/+* plants to analyse in their progeny if MpVHA-a and VHA-a3-a1-TD can functionally replace VHA-a1 at the TGN/EE. As we encountered T-DNA related silencing problems in earlier complementation experiments, we used CRISPR/Cas9 under control of an egg cell-specific promoter (*Wang et al., 2015*) to generate new *vha-a1* mutant alleles. Although *vha-a1* mutants were nearly indistinguishable from wildtype during vegetative growth they were found to be completely sterile due to a block in male gametophyte development. Given that we have previously reported that inducible knock-down of *VHA-a1* via either RNAi or amiRNA causes strong cell expansion defects (*Brüx et al., 2008*), the fact that plants homozygous for several independent null alleles are viable could be due to genetic compensation (*El-Brolosy and Stainier, 2017*). Indeed, when we systematically reduced the number of wildtype alleles for *VHA-a2* and *VHA-a3* in the *vha-a1* null background, a vegetative phenotype became manifest indicating that the two tonoplast-isoforms are able to complement for the lack of VHA-a1 during vegetative growth. It remains to be determined if *VHA-a2* or *VHA-a3* are up-regulated in the *vha-a1* null background, however we propose a model in which compensation is explained by rerouting of the tonoplast isoforms to theTGN/EE.

## VHA-a1 and VHA-a3 compete for entry into COPII vesicles

We assume that Marchantia reflects the ancestral state in which ER-export is mediated via COPII and acidification of the TGN/EE is either not required or is provided by V-ATPase complexes passing through en route to the tonoplast. Similar to yeast, the duplication of the ancestral VHA-a isoform provided the basis for a dedicated TGN/EE-isoform which in turn allowed the evolution of a Golgi-

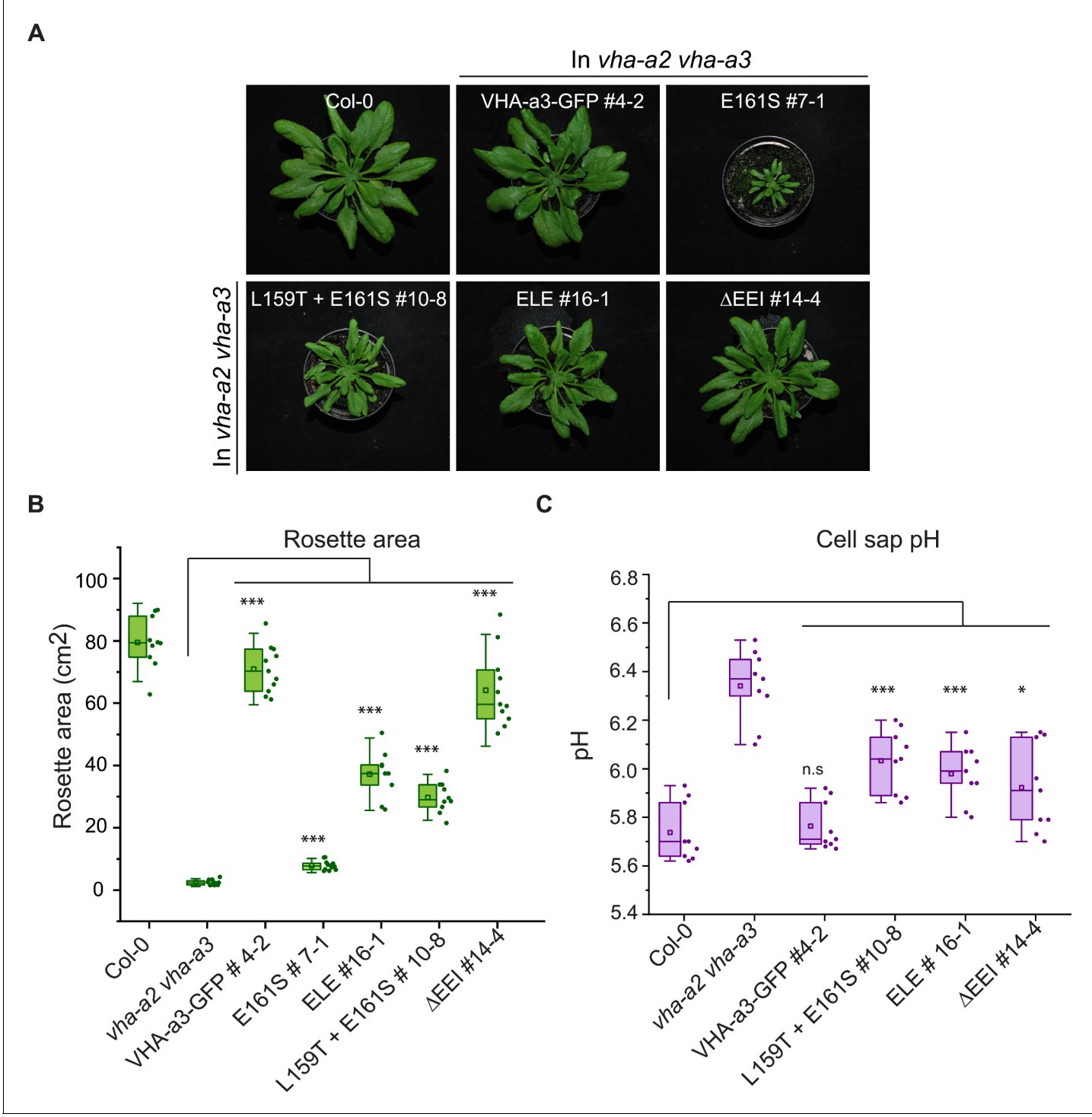

**Figure 7.** The mutated VHA-a1 subunits complement the *vha-a2 vha-a3* double mutant to varying degrees. Plants were grown in short day conditions (22 °C and 10 hr light) for 6 weeks. (**A**) All mutant variants of VHA-a1-GFP displayed bigger rosette size than the *vha-a2 vha-a3* double mutant. E161S which had a faint signal at the tonoplast in the *vha-a2 vha-a3* background also partially complemented the dwarf phenotype of the *vha-a2 vha-a3* double mutant. (**B**) Rosette area of 6-week-old plants grown under short day conditions. VHA-a1-GFP with ΔEEI mutation complements the *vha-a2 vha-a3* double mutant the best. (**C**) Mutated VHA-a1 containing V-ATPases have more alkaline cell sap pH values. Box plots center lines, medians; center boxes, means with n ≥ 9 measurements; box limits, 25th and 75th percentiles; whiskers extend to ±1.5 interquartile range. Asterisks indicate significant differences in the mean rosette area and cell sap pH. (Two-sample *t*-Test, p<0.05). (n.s: not significant, *p<0.05, **p<0.01 and ***p<0.001). The online version of this article includes the following source data and figure supplement(s) for figure 7:

**Source data 1.** Source data for *Figure 7B and C*.

*Figure 7 continued on next page*

*Figure 7 continued*

**Figure supplement 1.** The mutated VHA-a1-GFP proteins complement the *vha-a2 vha-a3* double mutant to varying degrees in long day conditions.
**Figure supplement 1—source data 1.** Source data for *Figure 7—figure supplement 1B*.
**Figure supplement 2.** Protein levels of the mutated VHA-a proteins at the tonoplast.
**Figure supplement 2—source data 1.** Source data for *Figure 7—figure supplement 2*.

independent trafficking route from the ER to the tonoplast, which we have shown previously to exist in meristematic root cells (*Viotti et al., 2013*). Alternatively, a single VHA-isoform could be able to enter both trafficking routes and it will thus be of importance to determine the evolutionary origin of the provacuolar route.

We propose here that in Arabidopsis, cells that require large quantities of tonoplast membranes such as the cells of the root tip, the VHA-a isoforms compete for entry into COPII vesicles. Assembly of all $V_O$-complexes takes place at the ER with the help of dedicated assembly factors and based on RNA expression and enzymatic activity the ratio of VHA-a1 to VHA-a3 is roughly 1:10 (*Hanitzsch et al., 2007*; *Neubert et al., 2008*). We assume that due to the presence of the a1-TD, the affinity of VHA-a1 for Sec24, the cargo receptor for COPII vesicles, is much higher than the affinity of VHA-a3. Small quantities of VHA-a3 can enter COPII vesicles but the majority is predominantly sorted into the Golgi-independent route (*Figure 11*).The fact that VHA-a3 is strongly trafficked to the TGN/EE by the insertion of the a1-TD provides further evidence for our competition model which also predicts that blocking of COPII-mediated export by DN-Sar1-GTP would lead to re-routing of VHA-a1 to the tonoplast when its affinity for COPII is reduced by point mutations in the a1-TD. The fact that tonoplast localization of a1-TD mutants is enhanced in the *vha-a2 vha-a3* background provides further support for a competition between the isoforms for the two ER-exits. Similarly, in the absence of VHA-a1, VHA-a2/VHA-a3 would be able to enter the COPII-route more efficiently and would thus be able to compensate for the lack of V-ATPases at the TGN/EE providing an explanation for the lack of a vegetative phenotype of the *vha-a1* mutant. We have shown previously that despite their seemingly strict spatial separation, the activities of the V-ATPase at the TGN/EE and the tonoplast are coordinated. In the *vha-a2 vha-a3* mutant in which VHA-a1 is the only remaining target, treatment with the V-ATPase inhibitor ConcA causes an additional increase in vacuolar pH (*Kriegel et al., 2015*) although the localization of VHA-a1 is not affected. Why then, does VHA-a3 not enter the COPII-route and rescue the reduced activity at the TGN/EE in the RNA-mediated knock-down lines of VHA-a1? Based on our model, we propose that in the RNA-mediated knock-down lines, although levels of VHA-a1 are reduced, they are still sufficient to outcompete VHA-a3 for entry into COPII vesicles.

In conclusion, our study provides the first report on a targeting motif for subunit a from a multicellular eukaryotic V-ATPase. Interaction partners that recognize and bind to the a1-TD need to be identified. ER-exit motifs as well as provacuole directing motifs in VHA-a3 should also be uncovered to better understand its trafficking via COPII vesicles and provacuoles. Our results incite further investigations to determine additional functions of the endosomal V-ATPase during vegetative growth and to learn what new functions it acquired during evolution that are essential for pollen development.

## Materials and methods

### Plant materials and growth conditions

*Arabidopsis thaliana*, Columbia 0 (Col-0) ecotype was used in all experiments in this study. The *vha-a2 vha-a3* double mutant was characterized by *Krebs et al., 2010*. *VHA-a1p:VHA-a1-GFP*, *VHA-a3p: VHA-a3-GFP* and *VHA-a1p:VHA-a1-RFP* lines were established by *Dettmer et al., 2006*. *BRI1-GFP* lines were previously established by *Geldner et al., 2007*.

Growth of Arabidopsis seedlings for confocal microscopy was performed on plates. The standard growth medium used contained 1/2 Murashige and Skoog (MS), 0.5% sucrose, 0.5% phyto agar, 10 mM MES and the pH was set to 5.8 using KOH. Agar and MS basal salt mixture were purchased from Duchefa. Seeds were surface sterilized with ethanol and stratified for 48 hr at 4°C. Plants were grown in long day conditions (LD; 16 hr light/8 hr dark) for 5 days.

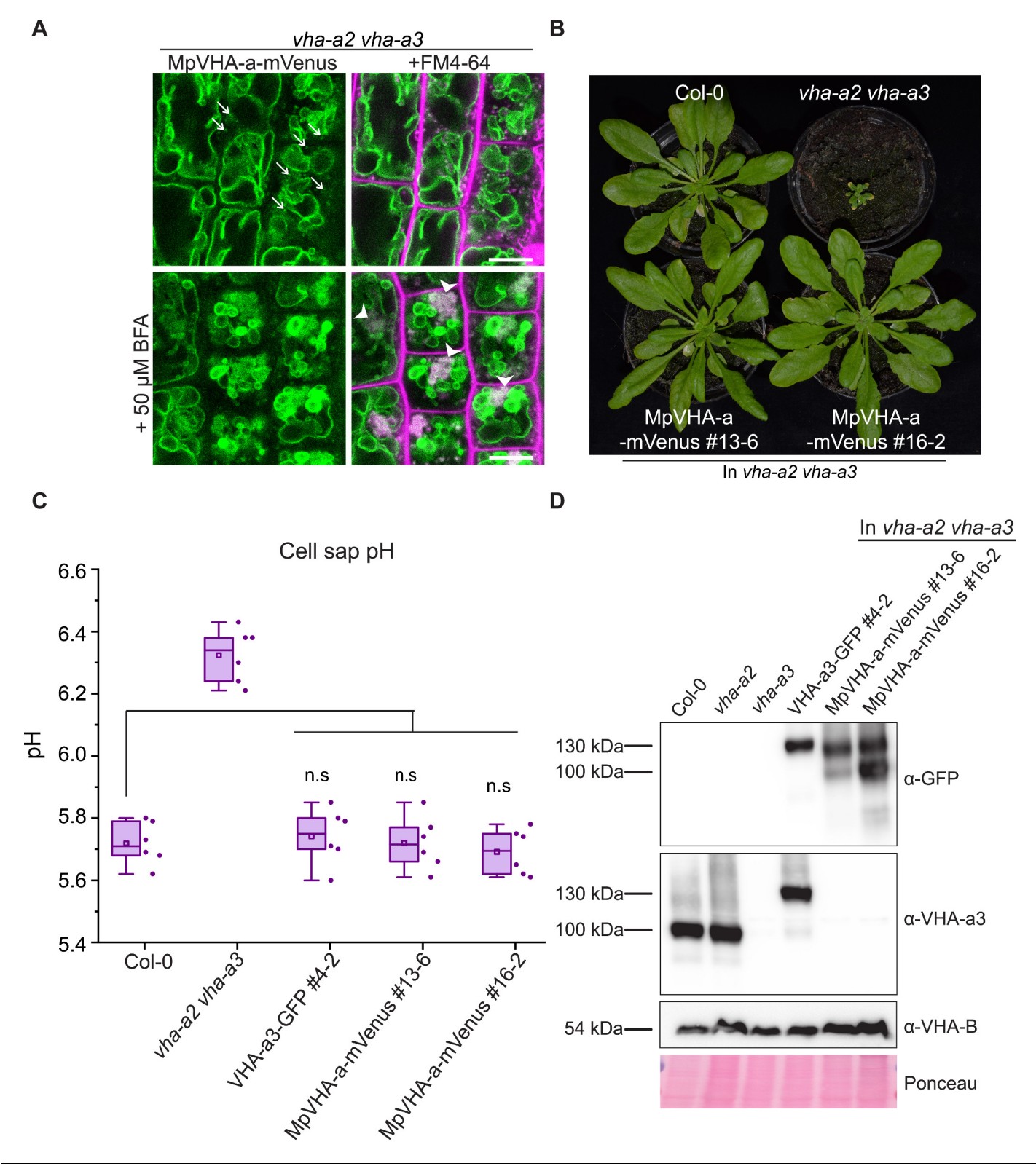

**Figure 8.** MpVHA-a containing complexes are functional in Arabidopsis and can replace VHA-a3 and VHA-a2 at the tonoplast. (**A**) MpVHA-a-mVenus is dual localized at the tonoplast and TGN/EE in the *vha-a2 vha-a3* background. The TGN/EE localization was confirmed by treatment of root cells with 50 µm BFA for 3 hr followed by staining with FM4-64 for 20 min. The core of BFA compartments were labeled with MpVHA-a-mVenus and FM4-64. Root tips of 6-day-old Arabidopsis seedlings were analyzed by CLSM. Green and magenta pseudo colors indicate fluorescence from GFP and FM4-64

*Figure 8 continued on next page*

*Figure 8 continued*

respectively. Scale bars = 10 μm. (B) MpVHA-a-mVenus can also fully complement the dwarf phenotype of the *vha-a2 vha-a3* double mutant. Plants for complementation assay were grown in SD conditions (22°C and 10 hr of light) for 6 weeks. (C) Cell sap pH of rosette leaves from plants grown in LD conditions (22°C and 16 hr of light) for 3 weeks. MpVHA-a-mVenus complexes restore the cell sap pH of the *vha-a2 vha-a3* double mutant to wildtype levels. Box plot center lines, medians; center boxes, means with n = 6 measurements from two biological replicates; box limits, 25th and 75th percentiles; whiskers extend to ±1.5 interquartile range. n.s indicates that there is no significant difference in the mean cell sap pH between the complementation lines and wildtype. (Two-sample *t*-Test, p<0.05). (D) Western blot of tonoplast membrane proteins from 6-week-old rosettes. Protein levels in the MpVHA-a-mVenus complementation lines is comparable to the VHA-a3-GFP complementation line.

The online version of this article includes the following source data and figure supplement(s) for figure 8:

**Source data 1.** Source data for *Figure 8C*.
**Source data 2.** Source data for *Figure 8D*.
**Figure supplement 1.** Subcellular localization of MpVHA-a-mVenus in *M. polymorpha*.
**Figure supplement 2.** MpVHA-a-mVenus is dual localized at the TGN/EE and tonoplast in Arabidopsis wildtype root cells.

For the rosette phenotype assays, seeds were stratified for 48 hr at 4°C and then placed on soil. Seedlings were transferred to individual pots at 7 days after germination (DAG). Plants were grown either in LD or short day conditions (SD; 10 hr light/14 hr dark) for the required time.

*M. polymorpha* accession Takaragaike-1 (Tak-1, male; *Ishizaki et al., 2008*) was used in this study. The growth condition and the transformation method were described previously (*Kubota et al., 2013*; *Kanazawa et al., 2016*). *M. polymorpha* expressing mRFP-MpSYP6A was generated in the previous work (*Kanazawa et al., 2016*).

## Construct design and plant transformation

### VHA-a1/VHA-a3 Chimeras

Five chimeric proteins were made which consisted of increasing lengths of the VHA-a1 N-terminus (37aa, 85aa, 131aa, 179 aa and 228 aa) fused to decreasing lengths of the C-terminal domain of VHA-a3. The GreenGate cloning system was used (*Lampropoulos et al., 2013*). Unique overhangs for each chimera were designed to allow seamless fusion of the *VHA-a1* and *VHA-a3* cDNA sequences (*Supplementary file 1*). The primers used are listed in *Supplementary file 1*. All PCR products were blunt end cloned into the pJET1.2 vector (ThermoFisher Scientific) and verified by sequencing (Eurofins). Verified clones were digested using either *Bgl*II *or Not*I *and Cla*I to release the fragments. To combine the *VHA-a1* and *VHA-a3* fragments, the GreenGate cloning system was applied using modules described in *Supplementary file 1*.

### UBQ:VHA-a3-a1-TD-GFP

The targeting domain of *VHA-a1* (*a1-TD*) was introduced into *VHA-a3* by PCR techniques. The N and C-termini of *VHA-a3* and the *a1-TD* were amplified using primers in *Supplementary file 1*. All PCR products were blunt end cloned into the pJET1.2 vector and verified by sequencing (Eurofins). Verified clones were digested using either *Bgl*II *or Not*I *and Cla*I to release the fragments. The destination vector was made by combining the fragments with GreenGate modules described in *Supplementary file 1*.

### Site-directed mutagenesis of VHA-a1

The pJET1.2 clone carrying VHA-a1 NT with 179 aa from the chimeras was used as a template for site-directed mutagenesis of VHA-a1. PCR mediated site-directed mutagenesis was performed using primers indicated in *Supplementary file 1* and according to *Wang and Malcolm, 1999*. Five cycles of single-primer extension reactions were carried out before the standard QuikChange Site-Directed Mutagenesis Protocol (Stratagene, La Jolla, CA, USA). Arabidopsis *VHA-a1 intron 10* was amplified from genomic DNA, a connecting *VHA-a1* fragment (*Y2*) and a C-terminal fragment (*VHA-a1-CT*) were amplified from *VHA-a1* cDNA using primers indicated in *Supplementary file 1*. All PCR fragments were subcloned into the pJet1.2 vector and were verified by sequencing (Eurofins). Verified clones were digested using *Bgl*II to release the fragments. The final destination vectors were made with modules described in *Supplementary file 1*.

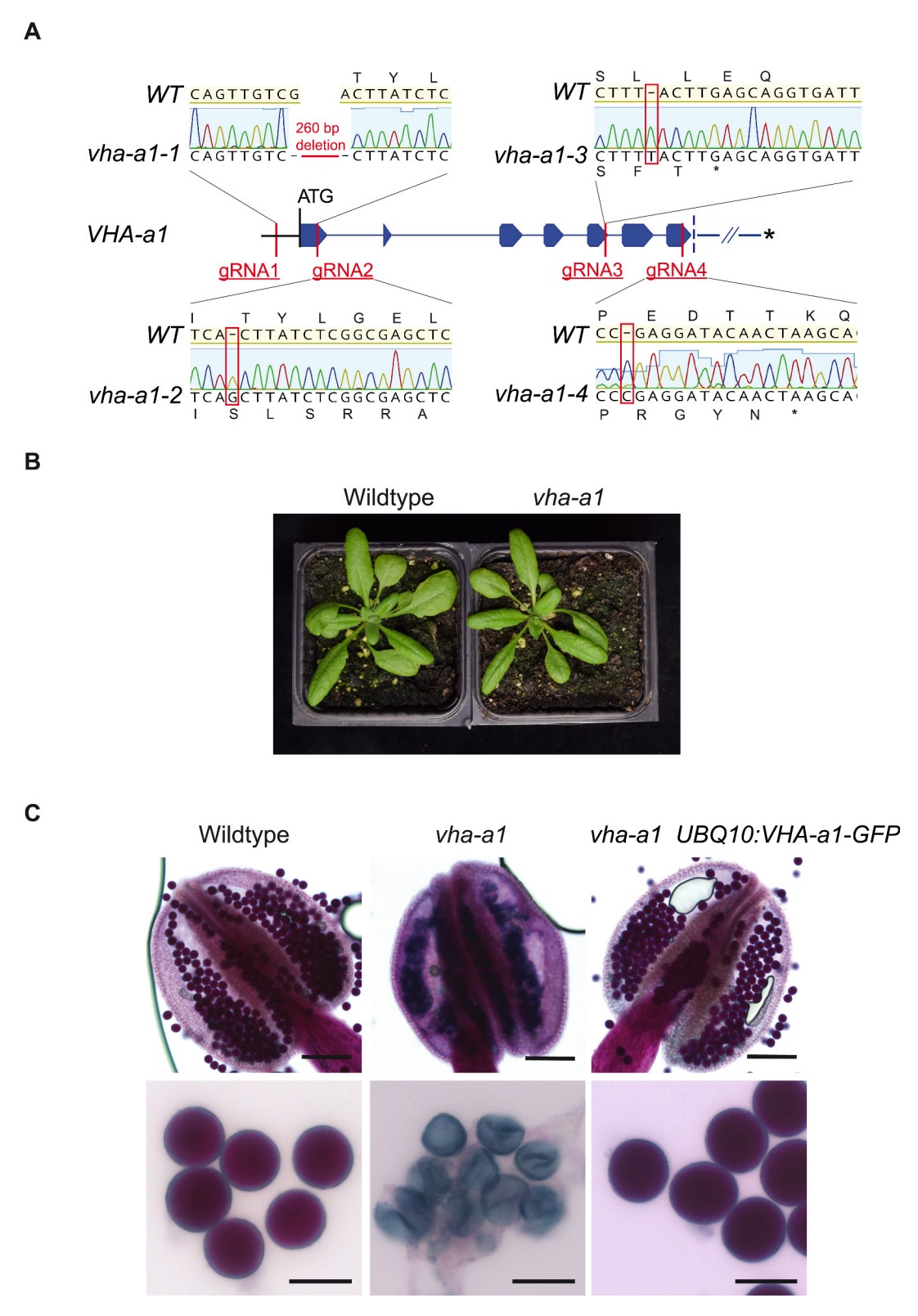

**Figure 9.** VHA-a1 is essential for pollen development but not for vegetative growth. (**A**) The exon-intron structure of the first seven exons of *VHA-a1* shows the sites which were targeted in independent CRISPR approaches. *vha-a1-1*, *vha-a1-2*, *vha-a1-3* and *vha-a1-4* are examples for *vha–a1* alleles that were obtained. (**B**) The *vha-a1* mutant is indistinguishable from wild type during vegetative growth. Plants were grown under long day conditions (3.5 weeks, 22°C and 16 hr of light). (**C**) Misshaped microspores/pollen in *vha-a1* anthers were visualized using Alexander's stain. VHA-a1-GFP rescues the
*Figure 9 continued on next page*

*Figure 9 continued*

pollen phenotype of *vha-a1*. VHA-a1-GFP is expressed under the *UBQ10* promoter. Scale bars, whole anthers: 100 μm, close ups of pollen grains: 20 μm.

The online version of this article includes the following figure supplement(s) for figure 9:

**Figure supplement 1.** *vha-a1-1* can be distinguished from the wildtype allele without sequencing and mutations in *VHA-a1-GFP* lead to absence of GFP signal.

**Figure supplement 2.** MpVHA-a-mVenus is dual localized in *vha-a1*.

## UBQ:VHA-a3R729N-GFP

To generate *UBQ:VHA-a3R729N-GFP*, the full length coding sequence of *VHA-a3* without Stop codon, was amplified from Arabidopsis Col-0 wild type cDNA using primer pairs a3*Sac*II-Fw and a3*Sal*I-Rv (*Supplementary file 1*). The resulting 2482 bp fragment was subcloned into pJET2.1blunt to receive *pJET1.2blunt-cVHA-a3-STOP*. Next, *pJET1.2blunt-cVHA-a3-STOP* was used as a template to perform site-directed mutagenesis PCR using primer combination cVHA-a3-R729N-Fw and cVHA-a3-R729N-Rv (*Supplementary file 1*) to receive *pJET1.2blunt-cVHA-a3R729N-STOP*. The coding sequence of *GFP5(S65T)* and an interconnecting linker was amplified from *VHA-a3-GFP* (*Dettmer et al., 2006*) with primer combination GFP5-*Sal*I-Fw2 and GFP5-*Pst*I-Rv2 (*Supplementary file 1*). The resulting 764 bp fragment was subcloned into pJET1.2blunt to receive *pJET1.2blunt-GFP(SalI/PstI)*. The coding sequences of *VHA-a3*, *GFP5* and the mutation R729N were verified by sequencing (Eurofins). To obtain the binary vector *UBQ10-VHA-a3-R729N-GFP*, *pUTKan* (*Krebs et al., 2012*) and *pJET1.2blunt-cVHA-a3R729N-STOP* were digested with *Sac*II and *Sal*I. The resulting 10847 and 2473 bp fragments were ligated to receive *UBQ10-VHA-a3-R729N*. Finally, *UBQ10-VHA-a3-R729N* and pJET1.2blunt-GFP(SalI/PstI) were digested with *Sal*I and *Pst*I. The 13291 and 756 bp fragments were ligated to obtain *UBQ10-VHA-a3-R729N-GFP*.

## Dex:Sar1BH74L-CFP

The Arabidopsis *Sar1B* cDNA sequence with the H74L mutation was synthesized by Eurofins. The synthesized fragment contained *Eco*31I at its 5' and 3' ends for subcloning. PCR was performed on the synthesized fragment using primers indicated in *Supplementary file 1* and the PCR product was blunt end cloned into the pJET1.2 vector. The *SarB1H74L* sequence was verified by sequencing (Eurofins). Verified clones were digested using *Not*I and *Cla*I to release the *SarB1H74L* fragments. The dexamethasone-inducible construct was made by using the LhG4/pOp system combined with the ligand-binding domain of the rat glucocorticoid receptor (GR; *Moore et al., 1998*; *Craft et al., 2005*; *Samalova et al., 2005*). Two GreenGate reactions were performed to create two intermediate vectors that were later combined on one T-DNA. The first intermediate vector (pKSM002) contained the GR-LhG4 transcription factor expressed under the *UBQ10* promoter and the second intermediate vector contained *Sar1BH74L* under the *pOP6* promoter (pKSN009). The two intermediate vectors were combined on one final destination vector pGGZ003 (*Supplementary file 1*).

## UBQ10:MpVHA-a-mVenus

The cDNA sequence of *MpVHA-a* was amplified by PCR using female *Marchantia polymorpha* (ecotype BoGa) thallus cDNA as a template. The sequence was amplified in two parts (*MpVHA-aNT* and *MpVHA-aCT)* separated at the exon 10-exon 11 junction using primers indicated in *Supplementary file 1*. All PCR products were blunt end cloned into the pJET1.2 vector and verified by sequencing (Eurofins). Verified clones were digested using *Bgl*II to release the fragments. The final destination vector was made with the *MpVHA-aNT*, *VHA-a1-intron10* and *MpVHA-aCT* fragments and modules described in *Supplementary file 1*.

## CaMV35S:MpVHA-a-mVenus and MpEF1α:MpVHA-a-mVenus

The genome sequence of *MpVHA-a* containing all exons and introns was amplified by PCR using the Tak-1 genome as a template with primers (*Supplementary file 1*). The amplified products were subcloned into pENTR D-TOPO (Invitrogen), and cDNA for *mVenus* was inserted into the *Asc*I site of the pENTR vectors using In-Fusion (Clontech) according to the manufacturer's instructions. The

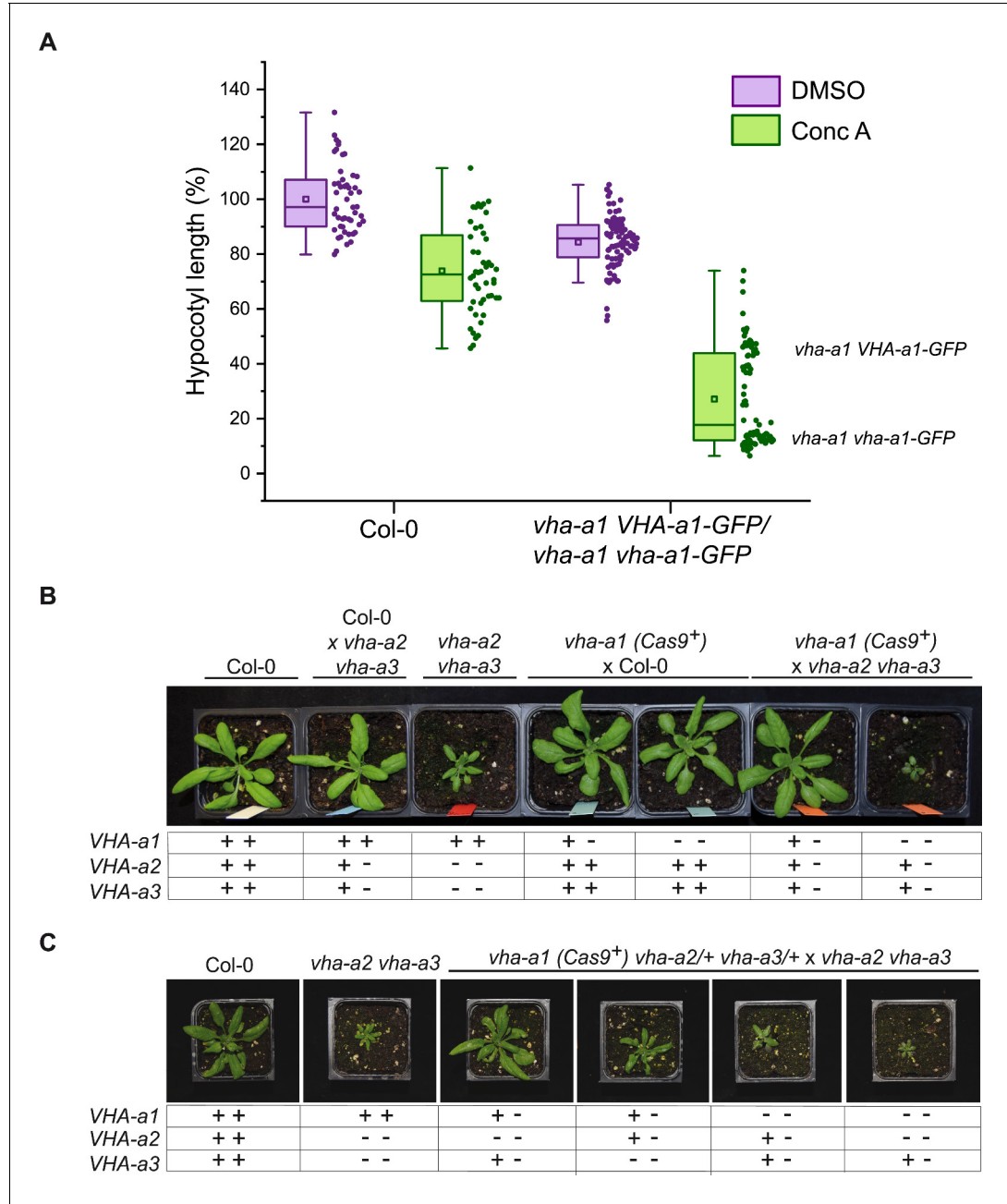

**Figure 10.** VHA-a2 and VHA-a3 replace VHA-a1 at the TGN/EE in *vha-a1* during vegetative growth. (**A**) *vha-a1 vha-a1*-GFP hypocotyls are hypersensitive to 125 nM ConcA, in contrast to wildtype hypocotyls indicating that a target of ConcA is present at the TGN/EE. *vha-a1* hypocotyls expressing *UBQ10:VHA-a1-GFP* are less sensitive to ConcA. Box plot center lines, medians; center boxes, means with n $\geq$ 48 measurements from three biological replicates; box limits, 25th and 75th percentiles; whiskers extend to ±1.5 interquartile range. (**B**) *vha-a1 (Cas9⁺)* was crossed with the *vha-a2 vha-a3* double mutant. Analysis of F1 plants revealed that *vha-a1 vha-a2/+ vha-a3/+* is reduced in growth. 3.5 week-old-plants are shown. (**C**) *vha-a1 (Cas9⁺) vha-a2/+ vha-a3/+* was crossed with the *vha–a2 vha–a3* double mutant. *vha-a1/+ vha-a2/+ vha-a3* is smaller than *vha-a1/+ vha-a2 vha-a3/+* and *vha–a1 vha–a2 vha–a3/+* was the smallest mutant found. 4 week-old-plants are shown. Plants were grown under long day conditions (22°C and 16 hr of light).

The online version of this article includes the following source data and figure supplement(s) for figure 10:

**Source data 1.** Source data for *Figure 10A*.
**Figure supplement 1.** *vha-a1* is hypersensitive to Concanamycin-A (ConcA).
**Figure supplement 1—source data 1.** Source data for *Figure 10—figure supplement 1A*.
**Figure supplement 2.** *vha-a1 vha-a2/+* and *vha-a1 vha-a3/+* have reduced rosette sizes.

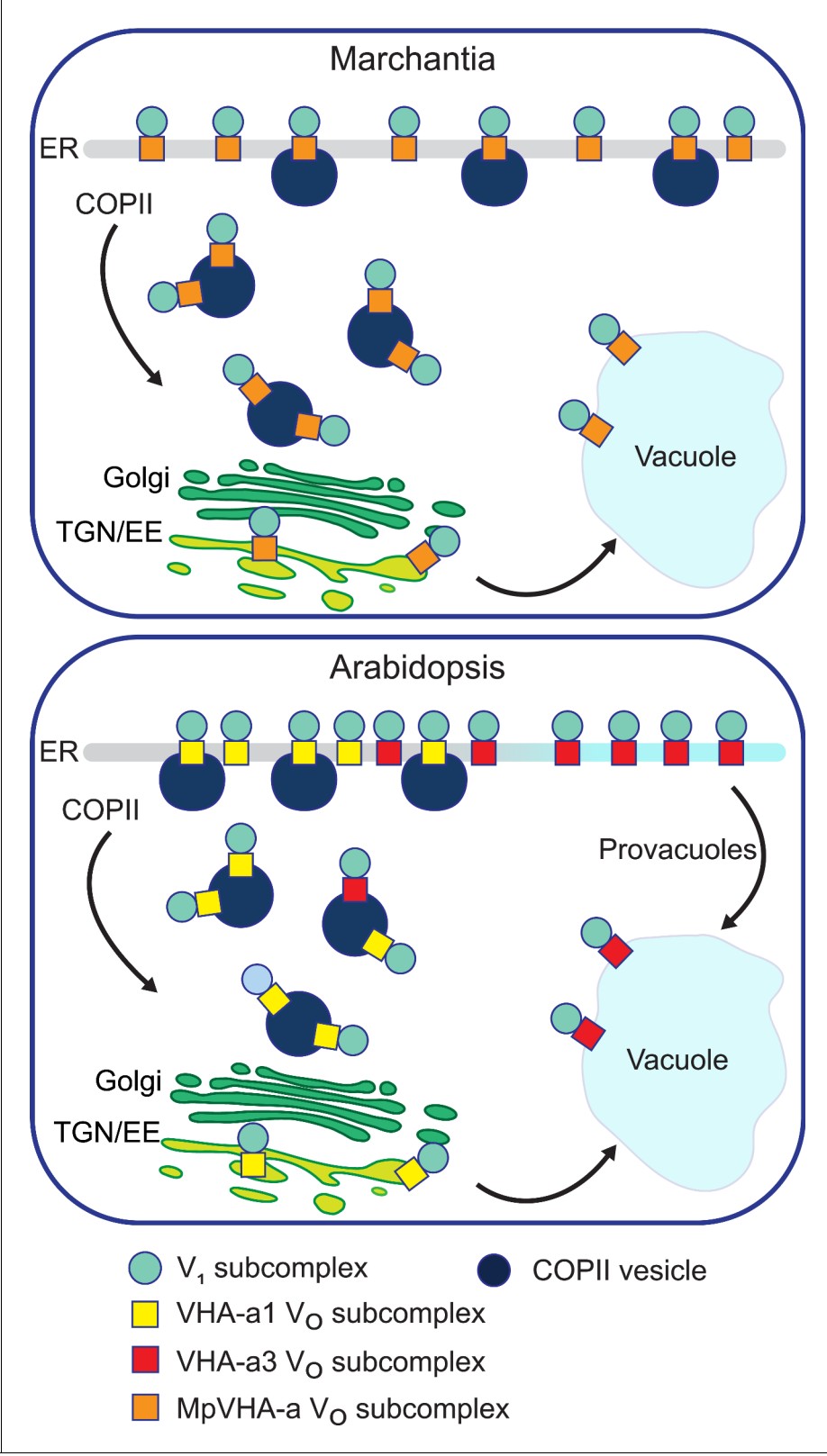

**Figure 11.** A competition exists to enter COPII vesicles in Arabidopsis. Upper panel: COPII-mediated ER-export is the only way to exit the ER in Marchantia. Marchantia VHA-a (MpVHA-a) does not contain a TGN/EE-retention signal. MpVHA-a complexes acidify the TGN/EE en route to the tonoplast. Lower panel: Two ER exits exist in Arabidopsis; COPII-mediated ER-export and exit via provacuoles. A competition exists between VHA-a1

*Figure 11 continued*

and VHA-a3/VHA-a2 complexes to enter COPII vesicles. VHA-a1 has a higher affinity for COPII machinery conferred by the a1-TD. Therefore, it is preferentially loaded into COPII vesicles over VHA-a3. VHA-a3 containing complexes are transported to the tonoplast via provacuoles.

resultant entry sequences were transferred to pMpGWB302 or pMpGWB303 (*Ishizaki et al., 2015*) using LR Clonase II (Invitrogen) according to the manufacturer's instructions.

## UBQ10:A.trichopoda-VHA-aNT-VHA-a1-mCherry,UBQ10:S.moellendorffii-VHA-aNT-VHA-a1-mCherry and UBQ10:P.taeda-VHA-aNT-VHA-a1-mVenus

The first 682 bp of the sequence identified as evm_27.model.AmTr_v1.0_scaffold00080.37 (*A. trichopoda VHA-a*), 649 bp of the sequence identified as 182335 (*S.moellendorffii-VHA-a*) and a 703 bp fragment starting from the 163rd bp of the sequence identified as 5A_I15_VO_L_1_T_29156/41278 (*P.taeda-VHA-a*) were synthesized (Eurofins). The synthesized fragments contained *Eco*31I at their 5' and 3' ends for subcloning. PCR was performed on the synthesized fragments using primers indicated in *Supplementary file 1* and the PCR products were blunt end cloned into the pJET1.2 vector. A connecting *VHA-a1* fragment (dodo) was also amplified from *VHA-a1* cDNA using primers in *Supplementary file 1*. All PCR fragments were verified by sequencing (Eurofins). Verified clones were digested using either *Bgl*II or *Not*I *and Cla*I to release the fragments. The destination vectors were made by combining the following components: *A.trichopoda-VHA-aNT* or *P.taeda-VHA-aNT* or *S.moellendorffii-VHA-aNT* with *VHA-a1 dodo*, *VHA-a1-intron10*, *VHA-a1-CT* and modules described in *Supplementary file 1*.

## UBQ10::ST-GFP

Rat *Sialyltransferase* (ST) sequence was amplified from pre-existing p16:ST-pHusion plasmid (*Luo et al., 2015*) using GG-ST-C-fwd and GG-ST-C-rev primers (*Supplementary file 1*), sub-cloned into pGGC000 after *Eco*31I digest and sequenced to verify correct insert sequence. pGGC-ST was used in a GreenGate reaction to generate *UBQ10:ST-GFP* using modules described in *Supplementary file 1*.

## UBQ10:VHA-a3-pmScarlet-I

The *VHA-a3* cDNA sequence was amplified from cDNA with primers listed in *Supplementary file 1*. After digest with *Eco*31I, VHA-a3 was subcloned into pGGC000. pGGC-VHA-a3 was used in a GreenGate reaction to assemble the final destination vector as described in *Supplementary file 1*.

## UBQ10:VHA-a1-GFP

The *VHA-a1* cDNA sequence was split into two parts by PCR to generate the fragments *VHA-a1 NT (Y5)* and *VHA-a1 CT* separated at the exon 10-exon 11 junction using primers indicated in *Supplementary file 1*. These fragments were digested with *Eco*31I and then subcloned into pGGC000 with *VHA-a1 intron 10*. pGGC-VHA-a1-intron 10 was sequenced and used in a GreenGate reaction with modules described in *Supplementary file 1* to generate the final destination vector.

## Destination vector pGGZ004

The vector backbone pGGZ004 is a GreenGate compatible plant binary vector which, unlike previous GreenGate vector backbones (*Lampropoulos et al., 2013*), contains the replicase required for plasmid replication in *A. tumefaciens* and therefore does not require the pSOUP helper plasmid (*Hellens et al., 2000*). pGGZ004 is based on the plant binary vector pTKan (*Krebs et al., 2012*), which originated from pPZP212 (*Hajdukiewicz et al., 1994*). To generate pGGZ004, pTKan was cut at its multiple cloning site with *Kpn*I followed by an incomplete digest with *Eco*31I, which opened the plasmid next to the T-DNA left border. The resulting 6971 bp fragment comprised the T-DNA border regions, the pBR322 *bom* site and the ColE1 and pVS1 plasmid origins for replication in *E. coli* and in *Agrobacterium*, respectively (*Hajdukiewicz et al., 1994*). GreenGate overhangs A and G were added to the opened vector backbone by sticky end ligation of annealed oligonucleotides *Eco*31I-w-AG-Fw and *Eco*31I-w-AG-Rv which contained *Kpn*I and *Eco*31I compatible overhangs. Lastly, a site-directed mutagenesis using primer combination pGGZ004SDM1-Fw and

pGGZ004SDM1-Rv was performed to remove an *Eco31*I site in the vector backbone (*Supplementary file 1*). The final pGGZ004 vector was fully sequenced before use for further cloning.

## CRISPR/Cas9 constructs

CRISPR target sites in *VHA-a1* were selected using CHOPCHOP (https://chopchop.cbu.uib.no; *Labun et al., 2016*) and CCtop; (https://crispr.cos.uni-heidelberg.de, *Stemmer et al., 2015*). gRNAs with at least 4 bp difference to every other region in the *Arabidopsis thaliana* genome were selected as precaution against off–target mutations. gRNA sequences (*Supplementary file 1*) were inserted into the plasmid pHEE401E as described by the authors (*Wang et al., 2015*). Three CRISPR plasmids were cloned, one containing two gRNAs (gRNA1 and gRNA2), aiming at deleting the region between the two CRISPR sites, and two plasmids containing one gRNA each, gRNA3 and gRNA4.

Depending on the final destination vector used, all constructs were transformed into either of two *Agrobacterium tumefaciens* strains. The strain GV3101:pMP90 was used if the final destination vector was pGGZ004. Selection was done on 5 mg/ml rifampicin, 10 mg/ml gentamycin, and 100 mg/ml spectinomycin. The strain ASE1(pSOUP+) was used if pGGZ001/3 were used. Selection was done on 100 µg/ml spectinomycin, 5 µg/ml tetracycline (for pSOUP), 25 µg/ml chloramphenicol, and 50 µg/ml kanamycin. Arabidopsis plants were transformed using standard procedures. Transgenic plants were selected on MS medium containing appropriate antibiotics.

## Analysis of mutations at CRISPR sites and genotyping of mutants

Genomic DNA was extracted from rosette leaves and amplified by PCR using primers flanking the CRISPR sites (*Supplementary file 1*). PCR products were sequenced by Eurofins or analyzed by agarose gel electrophoresis. Presence/absence of CRISPR T-DNA was monitored by PCR with *Cas9* specific primers (*Supplementary file 1*) followed by agarose gel electrophoresis. *VHA-a2* and *VHA–a3* wildtype and mutant alleles were identified by PCR using specific primers (*Supplementary file 1*) and agarose gel electrophoresis. Screening for *vha-a1-1* was done by detecting the PCR product spanning the CRISPR sites (260 bp shorter for *vha-a1-1* compared to wildtype) by agarose gel electrophoresis (*Figure 9—figure supplement 1*). Screening for *vha-a1* mutants with other *vha-a1* alleles was performed by observation of the pollen phenotype after having established that the defect in pollen development is caused by knockout of *VHA-a1* (*Supplementary file 1*).

## Pharmacological treatments and stains

Arabidopsis seedlings were incubated in liquid 1/2 MS medium with 0.5% sucrose, pH 5.8 (KOH), containing 50 µM BFA, 1 µM FM4-64, 60 µM DEX and 125 nM ConcA or the equivalent amount of DMSO in control samples for the required time at room temperature. Stock solutions were prepared in DMSO.

## Hypocotyl length measurement

For the hypocotyl length measurements, seeds were sown on agar plates made with distilled water without nutrients containing 2.5 mM MES, pH 5.8 adjusted with KOH, 0.8% phytoagar and 125 nM ConcA (Santa Cruz Biotechnology) dissolved in DMSO. After stratification for 48 hr at 4°C, a 4 hr light stimulus was given at 22°C. With the plates wrapped in aluminium foil the seedlings were grown in the dark for 4 days, then lain on overhead transparency sheets in drops of water and scanned. Quantification of hypocotyl length from the scans was done using Fiji (Fiji is just image J; *Schindelin et al., 2012*).

## Alexander staining for pollen viability

Anthers from flowers at stage 13 (*Smyth et al., 1990*) were incubated in Alexander's stain (*Alexander, 1969*) on objective slides covered with coverslips at room temperature for 20 hr. Bright field images were taken using a Zeiss Axio Imager M1 microscope. Adjustment of brightness and contrast of whole images was done with Fiji.

## Confocal microscopy image acquisition

Arabidopsis root cells of 6 day-old seedlings were analyzed by confocal laser scanning microscopy (CLSM) using a Leica TCS SP5II microscope equipped with a Leica HCX PL APO lambda blue 63.0 × 1.20 UV water immersion objective. Pictures were taken above the division zone. CFP, GFP and mVenus were excited at 458 nm, 488 nm and 514 nm with a VIS-argon laser respectively. mRFP, mCherry, pmScarlet-I and FM4-64 were excited at 561 nm with a VIS-DPSS 561 laser diode. For image acquisition, the Leica Application Suite Advanced Fluorescence software was used. Image post processing operations (Gaussian blur between 0.6 and 0.7; adjustment of brightness and contrast of the whole image) were done with Fiji.

CLSM on Marchantia thalli was performed according to *Kanazawa et al., 2016* with a minor modification. Briefly, the dorsal cells of 5-day-old thalli were observed using LSM780 confocal microscope (Carl Zeiss) equipped with an oil immersion lens (63×, N.A. = 1.4). For the spectral imaging, The samples were excited at 488 nm (Argon 488) and 561 nm (DPSS 561–10), and the emission was collected by 20 GaAsP detectors between 482 and 659 nm. Spectral unmixing and processing of the obtained images were conducted using ZEN2012 software (Carl Zeiss). For semi-superresolutional imaging, the samples were observed by LM780 confocal microscope equipped with Airyscan (Carl Zeiss). The samples were excited at 488 nm and 561 nm, and the emission was separated by BP420-480+BP495-550 (for mVenus), BP420-480+BP495-620 (for mRFP), and BP570-620+LP645 (for chlorophyll autofluorescence) filters, and detected by Airyscan. The semi-superresolutional processing was conducted using ZEN 2.3 SP1 software (Carl Zeiss).

## Image analysis and quantifications

### Colocalization analyses

Colocalization analysis was performed with the JACoP plugin on Image J (*Bolte, 2006*). Nine to ten images per condition were subjected to a mean blur of 2 pixels and a background subtraction with a rolling ball radius of 20 pixels. The Pearson correlation coefficient and the Mander's correlation coefficients M1 (VHA-a1-mRFP) and M2 (mutated VHA-a1-GFP) were determined (*Dunn et al., 2011*). For Mander's analysis a uniform threshold was set to all pictures to avoid background contribution. OriginPro was used to determine if the data were normally distributed (Shapiro-Wilk test) and to perform statistical tests. The data was found to be normally distributed and the student T-test was used to test for significance between means.

### Tonoplast signal intensity

Average tonoplast intensities were measured in roots of Arabidopsis seedlings expressing *DEX: Sar1BH74L-CFP* and *UBQ10:VHA-a1-GFP* with mutations or *VHA-a3:VHA-a3-GFP* after 6 hr induction with 60 µM DEX (Sigma-Aldrich). Images were acquired sequentially. In the first sequential scan, GFP was excited at 488 nm and emission detected at 500–557 nm. In the second sequential scan CFP was exited at 458 nm and emission detected between 468–481 nm. Intensity measurements were done using plot profiles in ImageJ. The maximum intensity values along line profiles across the TGN/EE and tonoplast were recorded. OriginPro was used to perform statistical tests. The data was found not to be normally distributed and the Mann-Whitney test was used to test for significance between means.

### Ratio of TGN/EE-to-tonoplast signal intensity

To determine the ratio of TGN/EE-to-tonoplast fluorescence intensity, images of root cells expressing UBQ10:VHA-a1-GFP with mutations in the wildtype and *vha-a2 vha-a3* background were acquired.The maximum intensity values along line profiles across the TGN/EE and tonoplast were measured using ImageJ. The average TGN/EE and tonoplast intensities were calculated for each image and the TGN/EE-to-tonoplast fluorescence intensity ratio was calculated. The TGN/EE-to-tonoplast fluorescence intensity ratios for $n \geq 10$ images for each mutation were averaged and statistical tests were performed with OriginPro. The student T-test was used to test for significance between means.

## High-pressure freezing, freeze substitution, and electronmicroscopy

Seven-day-old Arabidopsis wildtype seedlings expressing *DEX:AtSar1b-GTP* were induced for 6 hr with 60 µM DEX. Seedlings were then processed as previously described (*Scheuring et al., 2011*). Freeze substitution was performed in a Leica EM AFS2 freeze substitution unit in dry acetone supplemented with 0.3% uranyl acetate as previously described (*Hillmer et al., 2012*). Root tips were cut axially with a Leica Ultracut S microtome to obtain ultrathin sections. Sections were examined in a JEM1400 transmission electron microscope (JEOL) operating at 80 kV. Micrographs were recorded with a TemCam F416 digital camera (TVIPS, Gauting, Germany).

## pH measurements

Cell sap pH measurements were performed as previously described (*Krebs et al., 2010*).

## Tonoplast membrane preparation

Tonoplast membranes were prepared from rosette leaves of 6-week-old plants grown under short day conditions as previously described by *Barkla et al., 1999*; *Leidi et al., 2010*.

## SDS-PAGE and immunoblotting

Microsomal membrane and tonoplast membrane proteins were analyzed by SDS-PAGE and subsequent immunoblotting. Upon gel electrophoresis, the proteins were transferred to a PVDF membrane (Bio-Rad). The primary antibodies against VHA-a1 (AS142822) and VHA-a3 (AS204369) were purchased from Agrisera and were used in a dilution of 1:1000 in 2% BSA-TBS-T. The primary antibody against VHA-B (*Ward et al., 1992*) and anti-GFP (*Roth et al., 2018*) were previously described. Antigen on the membrane was visualized with horseradish peroxidase-coupled anti-rabbit IgG (Promega) for VHA-a1 and VHA-a3, anti-mouse IgG (Sigma) for VHA-B and chemiluminescent substrate (Peqlab). Immunostained bands were analyzed using a cooled CCD camera system (Intas).

## Phylogenetic analysis

For phylogenetic reconstruction in a first step the best molecular evolutionary model was determined by running the program PartitionFinder 2 (*Lanfear et al., 2017*). Phylogenetic reconstruction was then performed by running raxml-ng (*Kozlov et al., 2019*) setting the model to JTT+I+G and starting the analysis from 10 most parsimonious and 10 random trees. In order to estimate the reliability of the phylogenetic reconstruction 500 bootstrap replicates were run.

## Multiple sequence alignments

Multiple sequence alignments were performed using Clustal omega (*Madeira et al., 2019*). Aligned sequences were analyzed in Geneious 10.1.3.

## Homology modeling

3D models of the VHA-a1 and VHA-a3 N-termini were obtained through homology modeling with cryo-EM derived models of Stv1p-V$_O$ subcomplex (PDB6O7U) and Vph1p-V$_O$ subcomplex (PDB6O7T) as templates (*Vasanthakumar et al., 2019*). Homology modeling was performed according to *Roy et al., 2010*.

## Accession numbers

Sequence data from this article can be found in the Arabidopsis Genome Initiative or GenBank/ EMBL databases under the following accession numbers: VHA-a1, At2g28520; VHA-a3, At4g39080; Sar1B, AT1G56330.1. Data for *Marchantia polymorpha* VHA-a can be found on the Marchantia genome database with the following ID: Mp3g15140.1. *Amborella trichopoda* and *Selaginella moellendorffii* VHA-a sequences can be found on the Phytozome platform (*Goodstein et al., 2012*) with the following identifiers: *A.trichopoda* VHA-a; evm_27.TU.AmTr_v1.0_scaffold00080.37 and *S.moellendorffii* VHA-a; 182335. *Pinus taeda* sequence data can be found on the PineRefSeq project on the TreeGenes platform (*Falk et al., 2019*; *Wegrzyn et al., 2008*) with the following identifier: 5A_I15_VO_L_1_T_29156/41278.

## Acknowledgements

We would like to thank Prof. Dr. Sabine Zachgo (University of Osnabrück) for kindly providing us with *Marchantia polymorpha* (ecotype BoGa) cDNA. We also thank Prof. Dr. Stephan Rensing (University of Marburg) for providing us with *Selaginella moellendorffii* leaf tissue. We thank Dr. Rainer Waadt (COS, University of Heidelberg) for kindly providing GreenGate entry vectors. We thank Dr. Alyona Minina (COS, University of Heidelberg) for assisting with Alexander staining of pollen. We thank Dr. Takayuki Kohchi, Dr. Ryuichi Nishihama (Kyoto University), and Dr. Kimitsune Ishizaki (Kobe University) for vectors. We also thank Dr. Kazuo Ebine, Mayuko Yamamoto, Koji Hayashi (NIBB) and Fabian Fink (COS, University of Heidelberg) for supporting experiments. The support in *M. polymorpha* cultivation was provided by the Model Plant Research Facility of National Institute for Basic Biology. This work was supported by the Deutsche Forschungsgemeinschaft within SFB1101 to KS and by Grants-in-Aid for Scientific Research from the Ministry of Education, Culture, Sports, Science, and Technology of Japan (to TU, 19H05675, and 18H02470, and TK, 18K14738). Electron microscopy was performed by S Hillmer at the Electron Microscopy Core Facility of Heidelberg University with technical assistance of S Gold.

## Additional information

### Funding

| Funder | Grant reference number | Author |
| --- | --- | --- |
| DFG | SFB1101 TPA02 | Karin Schumacher |
| Ministry of Education, Culture, Sports, Science and Technology | 19H05675 | Takashi Ueda |
| Ministry of Education, Culture, Sports, Science and Technology | 18H02470 | Takashi Ueda |
| Ministry of Education, Culture, Sports, Science and Technology | 18K14738 | Takehiko Kanazawa |

The funders had no role in study design, data collection and interpretation, or the decision to submit the work for publication.

### Author contributions

Upendo Lupanga, Conceptualization, Investigation, Methodology, Writing - original draft; Rachel Röhrich, Investigation, Methodology, Writing - original draft; Jana Askani, Investigation, Methodology; Stefan Hilmer, Christiane Kiefer, Investigation; Melanie Krebs, Takashi Ueda, Writing - review and editing; Takehiko Kanazawa, Investigation, Writing - review and editing; Karin Schumacher, Conceptualization, Resources, Supervision, Funding acquisition, Investigation, Methodology, Writing - original draft, Project administration, Writing - review and editing

### Author ORCIDs

Upendo Lupanga https://orcid.org/0000-0001-7732-5012
Jana Askani http://orcid.org/0000-0002-1390-7344
Melanie Krebs https://orcid.org/0000-0001-6858-3247
Takashi Ueda http://orcid.org/0000-0002-5190-892X
Karin Schumacher https://orcid.org/0000-0001-6484-8105

### Decision letter and Author response

Decision letter https://doi.org/10.7554/eLife.60568.sa1
Author response https://doi.org/10.7554/eLife.60568.sa2

# Additional files

## Supplementary files

- Supplementary file 1. Supplementary tables.

- Transparent reporting form

## Data availability

All data generated or analysed during this study are included in the manuscript and supporting files. Source data is provided for Figures 2,4B, 5B, 7B and C, 8C and D, 10A, Supplemental Figure 8B, Supplemental Figure 9 and Supplemental Figure 15A.

The following previously published dataset was used:

| Author(s) | Year | Dataset title | Dataset URL | Database and Identifier |
|---|---|---|---|---|
| Jian Zhang, Xin-Xing Fu, Rui-Qi Li, Xiang Zhao, Yang Liu, Ming-He Li, Arthur Zwaenepoel, Hong Ma, Bernard Goffinet, Yan-Long Guan, Jia-Yu Xue, Yi-Ying Liao, Qing-Feng Wang, Qing-Hua Wang, Jie-Yu Wang, Guo-Qiang Zhang, Zhi-Wen Wang, Yu Jia, Mei-Zhi Wang, Shan-Shan Dong, Jian-Fen Yang, Yuan-Nian Jiao, Ya-Long Guo, Hong-Zhi Kong, An-Ming Lu, Huan-Ming Yang, Shou-Zhou Zhang, Yves Van de Peer, Zhong-Jian Liu & Zhi-Duan Chen | 2019 | The hornwort genome and early land plant evolution | https://doi.org/10.5061/dryad.msbcc2ftv | Dryad Digital Repository, 10.5061/dryad.msbcc2ftv |

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

# Appendix 1

**Appendix 1—key resources table**

| Reagent type (species) or resource | Designation | Source or reference | Identifiers | Additional information |
|---|---|---|---|---|
| Strain, strain background *Arabidopsis thaliana* | Col-0 | Nottingham Arabidopsis Stock center (NASC) | NASC: N37008 | |
| Strain, strain background (*Marchantia polymorpha*) | Tak-1 | *Ishizaki et al., 2008* PMID:18535011 | | |
| Genetic reagent *Arabidopsis thaliana* | *vha-a2 vha-a3* | (*Krebs et al., 2010*) PMID:20133698 | At2g21410,At4g39080 | SALK_142642 SALK_29786 |
| Genetic reagent *Arabidopsis thaliana* | *vha-a1-1* | this study | At2g28520 | See Materials and methods, Construct design and plant transformation |
| Genetic reagent *Arabidopsis thaliana* | *vha-a1-2* | this study | At2g28520 | See Materials and methods, Construct design and plant transformation |
| Genetic reagent *Arabidopsis thaliana* | *vha-a1-3* | this study | At2g28520 | See Materials and methods, Construct design and plant transformation |
| Genetic reagent *Arabidopsis thaliana* | *vha-a1-4* | this study | At2g28520 | See Materials and methods, Construct design and plant transformation |
| Genetic reagent *Arabidopsis thaliana* | BRI1: BRI-GFP | (*Geldner et al., 2007*) PMID:17578906 | | |
| Genetic reagent (*Marchantia polymorpha*) | CaMV35S: mRFPMpSyp6A | (*Kanazawa et al., 2016*) PMID:26019268 | | |
| Gene *Arabidopsis thaliana* | VHA-a1 | arabidopsis.org | At2g28520 | |
| Gene *Arabidopsis thaliana* | VHA-a3 | arabidopsis.org | At4g39080 | |
| Gene *Arabidopsis thaliana* | Sar1B | arabidopsis.org | At1g56330 | |
| Gene (*Rattus norvegicus*) | Beta-galactoside alpha-2,6-sialyltransferase 1 (ST) | RGD | 3676 | |
| Gene *Arabidopsis thaliana* | Brassinosteroid insensitive 1 (BRI1) | arabidopsis.org | At4g39400 | |
| Gene (*Marchantia polymorpha*) | MpVHA-a | Marchantia genome database | Mp3g15140 | |

*Continued on next page*

*Appendix 1—key resources table continued*

| Reagent type (species) or resource | Designation | Source or reference | Identifiers | Additional information |
|---|---|---|---|---|
| Gene *Arabidopsis thaliana* | MpSYP6A | Marchantia genome database | Mp3g18380 | |
| Gene (*Amborella trichopoda*) | *Amborella trichopoda VHA-a* | Phytozome | evm_27.TU.AmTr _v1.0_scaffold00080.37 | |
| Gene (*Selaginella moellendorffii*) | *Selaginella moellendorffii VHA-a* | Phytozome | 182335 | |
| Gene (*Pinus taeda*) | *Pinus taeda VHA-a* | PineRefSeq, Tree genes | 5A_I15_VO_L_1_ T_29156/41278 | |
| Transfected construct *Arabidopsis thaliana* | CRISPR VHA-a1 U6-26p: gRNA one and U6-29p:gRNA two in pHEE401E | this study | | Materials and methods, Construct design and plant transformation |
| Transfected construct *Arabidopsis thaliana* | CRISPR VHA-a1 U6-26p: gRNA three in pHEE401E | this study | | Materials and methods, Construct design and plant transformation |
| Transfected construct *Arabidopsis thaliana* | CRISPR VHA-a1 U6-26p: gRNA four in pHEE401E | this study | | Materials and methods, Construct design and plant transformation |
| Transfected construct *Arabidopsis thaliana* | UBQ10: VHA-a1 NT 35 aa-VHA-a3 | this study | | See Materials and methods, Construct design and plant transformation |
| Transfected construct *Arabidopsis thaliana* | UBQ10: VHA-a1 NT 85 aa-VHA-a3 | this study | | See Materials and methods, Construct design and plant transformation |
| Transfected construct *Arabidopsis thaliana* | UBQ10: VHA-a1 NT 131 aa-VHA-a3 | this study | | See Materials and methods, Construct design and plant transformation |
| Transfected construct *Arabidopsis thaliana* | UBQ10: VHA-a1 NT 179 aa-VHA-a3 | this study | | See Materials and methods, Construct design and plant transformation |
| Transfected construct *Arabidopsis thaliana* | UBQ10: VHA-a1 NT 228 aa-VHA-a3 | this study | | See Materials and methods, Construct design and plant transformation |
| Transfected construct *Arabidopsis thaliana* | UBQ10:VHA-a1-intron10-GFP | this study | | See Materials and methods, Construct design and plant transformation |
| Transfected construct *Arabidopsis thaliana* | UBQ:VHA-a3-pmScarlet-I | this study | | See Materials and methods, Construct design and plant transformation |
| Transfected construct *Arabidopsis thaliana* | UBQ10:VHA-a1-intron10 E156Q-GFP | this study | | See Materials and methods, Construct design and plant transformation |

*Continued on next page*

*Appendix 1—key resources table continued*

| Reagent type (species) or resource | Designation | Source or reference | Identifiers | Additional information |
|---|---|---|---|---|
| Transfected construct *Arabidopsis thaliana* | UBQ10:VHA-a1-intron10 E161S-GFP | this study | | See Materials and methods, Construct design and plant transformation |
| Transfected construct *Arabidopsis thaliana* | UBQ10:VHA-a1-intron10 F134Y-GFP | this study | | See Materials and methods, Construct design and plant transformation |
| Transfected construct *Arabidopsis thaliana* | UBQ10:VHA-a1-intron10 L159T-GFP | this study | | See Materials and methods, Construct design and plant transformation |
| Transfected construct *Arabidopsis thaliana* | UBQ10:VHA-a1-intron10 E156Q + L159T-GFP | this study | | See Materials and methods, Construct design and plant transformation |
| Transfected construct *Arabidopsis thaliana* | UBQ10:VHA-a1-intron10 L159T + E161S -GFP | this study | | See Materials and methods, Construct design and plant transformation |
| Transfected construct *Arabidopsis thaliana* | UBQ10:VHA-a1-intron10 ELE-GFP | this study | | See Materials and methods, Construct design and plant transformation |
| Transfected construct *Arabidopsis thaliana* | UBQ10:VHA-a1-intron10 ΔEEI-GFP | this study | | See Materials and methods, Construct design and plant transformation |
| Transfected construct *Arabidopsis thaliana* | UBQ:VHA-a3R729N-GFP | this study | | See Materials and methods, Construct design and plant transformation |
| Transfected construct *Arabidopsis thaliana* | UBQ10:VHA-a3-a1-TD-GFP | this study | | See Materials and methods, Construct design and plant transformation |
| Transfected construct *Arabidopsis thaliana* | Dex:Sar1BH74L-CFP | this study | | See Materials and methods, Construct design and plant transformation |
| Transfected construct *Arabidopsis thaliana* | UBQ10:MpVHA-a-mVenus | this study | | See Materials and methods, Construct design and plant transformation |
| Transfected construct (*Marchantia polymorpha*) | CaMV35S:MpVHA-a-mVenus | this study | | See Materials and methods, Construct design and plant transformation |
| Transfected construct (*Marchantia polymorpha*) | MpEF1α:MpVHA-a-mVenus | this study | | See Materials and methods, Construct design and plant transformation |
| Transfected construct *Arabidopsis thaliana* | UBQ: A.trichopoda -VHA-a-NT-VHA-a1-mCherry | this study | | See Materials and methods, Construct design and plant transformation |

*Continued on next page*

*Appendix 1—key resources table continued*

| Reagent type (species) or resource | Designation | Source or reference | Identifiers | Additional information |
|---|---|---|---|---|
| Transfected construct *Arabidopsis thaliana* | UBQ:P.taeda-VHA-a-NT-VHA-a1-mVenus | this study | | See Materials and methods, Construct design and plant transformation |
| Transfected construct *Arabidopsis thaliana* | UBQ:S. moellendorffii -VHA-a-NT-VHA-a1-mCherry | this study | | See Materials and methods, Construct design and plant transformation |
| Transfected construct *Arabidopsis thaliana* | UBQ:ST-GFP | this study | | See Materials and methods, Construct design and plant transformation |
| Recombinant DNA reagent (plasmid) | UBQ10 promoter | *Lampropoulos et al., 2013* PMID:24376629 | pGGA006 | |
| Recombinant DNA reagent (plasmid) | pOp6 | *Schürholz et al., 2018* PMID:30026289 | pGGA016 | |
| Recombinant DNA reagent (plasmid) | B-Dummy | *Lampropoulos et al., 2013* PMID:24376629 | pGGB003 | |
| Recombinant DNA reagent (plasmid) | entry module C | *Lampropoulos et al., 2013* PMID:24376629 | pGGC000 | |
| Recombinant DNA reagent (plasmid) | pGGC-VHA-a3 | this study | pKSC003 | See Materials and methods, Construct design and plant transformation |
| Recombinant DNA reagent (plasmid) | pGGC-VHA-a1-intron 10 | this study | pKSC012 | See Materials and methods, Construct design and plant transformation |
| Recombinant DNA reagent (plasmid) | pGGC-ST | this study | pKSC013 | See Materials and methods, Construct design and plant transformation |
| Recombinant DNA reagent (plasmid) | GR-LhG4 | *Schürholz et al., 2018* PMID:30026289 | pGGC018 | |
| Recombinant DNA reagent (plasmid) | linker GFP | *Lampropoulos et al., 2013* PMID:24376629 | pGGD001 | |
| Recombinant DNA reagent (plasmid) | linker-CFP | *Lampropoulos et al., 2013* PMID:24376629 | pGGD004 | |
| Recombinant DNA reagent (plasmid) | GFP (A206K) no linker | *Lampropoulos et al., 2013* PMID:24376629 | pGGD011 | |
| Recombinant DNA reagent (plasmid) | GSL-mVenus | this study | p2456 | See Materials and methods, Construct design and plant transformation |
| Recombinant DNA reagent(plasmid) | GSL-pmScarlet-I | this study | p1324 | See Materials and methods, Construct design and plant transformation |

*Continued on next page*

*Appendix 1—key resources table continued*

| Reagent type (species) or resource | Designation | Source or reference | Identifiers | Additional information |
|---|---|---|---|---|
| Recombinant DNA reagent (plasmid) | GSL-mCherry | *Waadt et al., 2017* PMID:28850185 | p2897 | |
| Recombinant DNA reagent (plasmid) | rbcS terminator | *Lampropoulos et al., 2013* PMID:24376629 | pGGE001 | |
| Recombinant DNA reagent (plasmid) | HSP18.2M terminator | *Waadt et al., 2017* PMID:28850185 | p1296 | |
| Recombinant DNA reagent (plasmid) | BastaR | *Lampropoulos et al., 2013* PMID:24376629 | pGGF001 | |
| Recombinant DNA reagent (plasmid) | SulfR | *Lampropoulos et al., 2013* PMID:24376629 | pGGF012 | |
| Recombinant DNA reagent (plasmid) | KanR | *Lampropoulos et al., 2013* PMID:24376629 | pGGF007 | |
| Recombinant DNA reagent (plasmid) | HygR_pNos | *Waadt et al., 2017* PMID:28850185 | p1317 | |
| Recombinant DNA reagent (plasmid) | HygR_pUbq10 | *Lampropoulos et al., 2013* PMID:24376629 | pGGF005 | |
| Recombinant DNA reagent (plasmid) | F-H adapter | *Lampropoulos et al., 2013* PMID:24376629 | pGGG001 | |
| Recombinant DNA reagent (plasmid) | H-A adapter | *Lampropoulos et al., 2013* PMID:24376629 | pGGG002 | |
| Recombinant DNA reagent (plasmid) | intermediate vector M | *Lampropoulos et al., 2013* PMID:24376629 | pGGM000 | |
| Recombinant DNA reagent (plasmid) | intermediate vector N | *Lampropoulos et al., 2013* PMID:24376629 | pGGN000 | |
| Recombinant DNA reagent (plasmid) | UBQ10:GR-LHG4: trbcS | this study | pKSM002 | See Materials and methods, Construct design and plant transformation |
| Recombinant DNA reagent (plasmid) | pOP6:Sar1BH74L-CFP | this study | pKSN009 | See Materials and methods, Construct design and plant transformation |
| Recombinant DNA reagent (plasmid) | pGGZ001 | *Lampropoulos et al., 2013* PMID:24376629 | | |
| Recombinant DNA reagent (plasmid) | pGGZ003 | *Lampropoulos et al., 2013* PMID:24376629 | | |
| Recombinant DNA reagent (plasmid) | pGGZ004 | this study | | See Materials and methods, Construct design and plant transformation |
| Recombinant DNA reagent (plasmid) | pHEE401E | *Wang et al., 2015* PMID:26193878 | | See Materials and methods, Construct design and plant transformation |

*Continued on next page*

*Appendix 1—key resources table continued*

| Reagent type (species) or resource | Designation | Source or reference | Identifiers | Additional information |
|---|---|---|---|---|
| Antibody | VHA-a1 (rabbit polyclonal) | Agrisera | AS142822 | (1:1000) |
| Antibody | VHA-a3 (rabbit polyclonal) | Agrisera | AS204369 | (1:1000) |
| Antibody | VHA-B (mouse polyclonal) | *Ward et al., 1992* PMID:16668845 | | (1:100) |
| Antibody | anti-GFP (rabbit polyclonal) | *Roth et al., 2018* PMID:30410018 | | (1:5000) |
| Chemical compound, drug | Concanamycin-A (ConcA) | Santa Cruz | sc-202111A | |
| Chemical compound, drug | Brefeldin A | LC Laboratories | B-8500 | |
| Chemical compound, drug | FM4-64 | Thermo Fisher Scientific | T13320 | |
| Chemical compound, drug | Dexamethasone (DEX) | Sigma-Aldrich | D4902 | |
| Chemical compound, drug | Malachite Green, Acid Fuchsin, Orange G | Thermo Fisher Scientific | AC413490250, AC400210250, AC229820250 | Alexander stain |
| Commercial assay or kit | CloneJET PCR Cloning kit | Thermo Fisher Scientific | K1231 | |
| Strain, strain background (*E. coli*) | NEBα | In-house facility | | COS Heidelberg |
| Strain, strain background (*A. tumefaciens*) | GV3101 | In-house facility | | COS Heidelberg |
| Strain, strain background (*A. tumefaciens*) | ASE1 | In-house facility | | COS Heidelberg |
| Software, algorithm | Adobe Illustrator 2020 | Adobe Inc | | Figure assembly |
| Software, algorithm | Zen Software | Carl Zeiss | | Microscopy |
| Software, algorithm | Leica LSF | Leica | | Microscopy |
| Software, algorithm | Originpro 2020 | Origin | | Statistics and graph plotting |
| Software, algorithm | Intas Imager | Intas | | Western blot |
| Software, algorithm | Image J | NIH | | Image quantification |

# Appendix 2

## Legend to *Supplementary file 1*

| | |
|---|---|
| Table 1 | Statistical support for the predicted structural homology models obtained from I-TASSER |
| Table 2 | Overview of the number of VHA-a isoforms in plants with VHA-a isoforms that do not contain the a1-TD |
| Table 3 | Overview T1 analysis CRISPR/Cas9 VHA-a1 lines |
| Table 4 | Segregation of *vha-a1-1* |
| Table 5 | Chimeric proteins overhangs and primers |
| Table 6 | GreenGate modules and constructs |
| Table 7 | CRISPR/Cas9 gRNA sequences and genotyping primers |

