## [Decision Letter]

**Acceptance summary:**

Vacuolar-type H^+^ATPases (V-ATPases) are multiprotein complexes that contribute to acidification of the secretory pathway and vacuole. This paper employs a sophisticated combination cell biology and evolutionary analyses to probe the molecular mechanisms by which V-ATPases are targeted within plant cells. The authors determine the signal for complex targeting within VHAa subunits and propose a model in which two separate trafficking mechanisms at the ER might compete to target V-ATPases complexes containing different VHAa subunits to the vacuole or secretory pathway.

**Decision letter after peer review:**

Thank you for submitting your article "The Arabidopsis V-ATPase is localized to the TGN/EE via a seed plant specific motif" for consideration by *eLife*. Your article has been reviewed by three peer reviewers, one of whom is a member of our Board of Reviewing Editors, and the evaluation has been overseen by Christian Hardtke as the Senior Editor. The following individual involved in review of your submission has agreed to reveal their identity: Farhah Assaad (Reviewer #2).

The reviewers have discussed the reviews with one another and the Reviewing Editor has drafted this decision to help you prepare a revised submission.

Summary:

This manuscript describes how the V-ATPase VHA-a1 is retained at the Arabidopsis TGN/EE through an N-terminal motif that is specific to seed plants. Using domain-swaps, site-directed mutagenesis, and live-cell imaging of fluorescently tagged constructs, the authors define a protein motif required for ER exit and TGN retention of VHA-a1 in Arabidopsis. The authors show that the TGN/EE-retention motif is limited to seed plants, and elegantly demonstrate that this region from other seed plants is capable of directing chimeric VHA-a1 to the TGN/EE in Arabidopsis, whereas this region from a non-seed plant is not. Finally, the authors identify a specific requirement for VHA-a1 in pollen development that cannot be complemented by tonoplast-localised VHA-a2, VHA-a3 or the single Marchantia V-ATPase. This suggests a previously undescribed role for VHA-a1 in pollen development that has arisen in seed plants.

Essential revisions:

The three reviewers are in agreement that the manuscript is of high quality and general interest, but that there are some revisions that could substantially improve the manuscript:

1) Quantification should be performed of the colocalization analyses.

2) The "punctate localization" of MpVHAa in Marchantia is not convincing as presented, so either additional experiments or substantial rewording of the text and conclusions are required.

3) Small, but important, clarifications to the text, figures and figure legends, and methods, are required throughout, as detailed in each review.

Reviewer #1:

Like other organisms, in Arabidopsis, the V-type ATPase is targeted to different compartments based on motifs in the N-terminus of the "a" subunit (VHAa1 to the TGN/early endosome, a2 and a3 to the tonoplast), however the exact signals have not been defined in plants. Using domain-swaps, site-directed mutagenesis, and live-cell imaging of fluorescently tagged constructs, the authors define a region of VHAa1 that is necessary for VHAa1 TGN localization (Figure 2) and sufficient to partially relocalize VHAa3 to the TGN (Figure 1). Using an inducible, dominant-negative construct affecting ER export, they demonstrate that TGN localization of VHAa1, but not tonoplast targeting, is dependent upon ER-to-Golgi COPII vesicle trafficking (Figures 3-5). Interestingly, partial mislocalization of VHAa1 from the TGN to the tonoplast was correlated with rescue of vhaa2/vhaa3 phenotypes, implying that their localization is different, but their biochemical function remains conserved (Figure 7). However, complementation experiments with Marchantia VHAa demonstrate that while it can localize to both the tonoplast and TGN (Figure 8), it can complement vhaa2/vhaa3 phenotypes, but not vhaa1 in pollen development (Figure 9).

The authors also investigate the evolution of the VHAa1 targeting motif. Sequence alignments and localization experiments demonstrate the seed plant VHAa1 and VHAa3 cluster into distinct domains, that the VHAa1 targeting sequence is conserved and that the targeting domain of VHAa1 from gymnosperm (pine) and the basal flowering plant *A. trichopoda* are sufficient to localize AtVHAa1 TGN in Arabidopsis. However, green plants outside the spermatophytes seem to lack this distinction (Figure 6). The authors propose a model in which two separate mechanisms at the ER might compete to target V-ATPase complexes containing VHAa3 to the tonoplast (via a COP-II independent mechanism) and VHAa1 to the TGN (Figure 10).

This paper has a nice combination of mechanistic and evolutionary questions and the model they propose seems internally consistent. The manuscript is written clearly, but it would benefit from some additional quantitative analyses.

1) Please conduct quantitative colocalization analysis of micrographs. Two freely available and well-documented plugins for Fiji/ImageJ are DiAna (Gilles et al. 2017, Methods) and JaCOP (Bolte and Cordelières 2006, J Microscopy).

2) Tonoplast morphology seems quite different in some of the VHAa1-TD point mutants (particularly ELE in Figure 4); please clarify how "tonoplast" signal was defined for the quantification in Figure 4B. Also, what is the magenta signal in Figure 4A? This is not clear from the labels or figure legend and it does not seem to display the same post-DEX behaviour as either VHAa1 or SAR1bGTP as presented in Figure 3. If it is VHAa1-RFP signal, the data in Figure 4B (and 5B) might be better presented as a decrease in colocalization between VHAa1 point mutants and VHAa1-RFP (or FM4-64 in Figure 5), corrected for fluorescence intensity, since this might be clearer than trying to define which pixels belong to the tonoplast.

3) I cannot see any colocalization between MpVHA-a-mVenus and mRFP-MpSYP6A in Figure 8—figure supplement 1. Please either revise the discussion of these images, or provide more compelling evidence of this colocalization (better images, quantification, treatment with BFA, if this works in Marchantia?)

4) It seems that the data presented in Figure 7C does not include any biological replication since only three points are plotted and the legend clearly indicates "n=3 technical replicates". Please perform additional biological replicates for this experiment.

5) Are VHAa2 or VHAa3 localized to the TGN in vegetative tissue of the vhaa1 mutants? This would provide a second line of evidence to the authors' genetic results that VHAa2/VHAa3 can replace VHAa1 during vegetative growth and it might help to clarify the differences observed between VHAa1 RNAi lines and VHAa1 CRISPR mutants.

6) The model in Figure 10 illustrates a kind of competition between ER trafficking components for V-ATPases containing VHAa1 for TGN targeting via a COPII-dependent mechanism and VHAa3 for tonoplast targeting via a COPII-independent mechanism. Since VHAa2/a3 cannot compensate for loss of VHAa1 in pollen development, then this model predicts that overexpression of VHAa2/a3 could potentially cause a pollen development phenotype by displacing some VHAa1 at the TGN. Similarly, the model predicts reduced trafficking of VHAa1-TD point mutants (or VHAa3-a1-TD) to the tonoplast when VHAa2/a3 are overexpressed. Have these ever been observed (e.g. in VHAa3-GFP lines, or with VHAa3-GFP in the VHAa1 RNAi line)?

Reviewer #2:

This manuscript describes how the V-ATPase VHA-a1 is retained at the Arabidopsis TGN/EE through an N-terminal motif that is specific to seed plants. The authors are able to substantially narrow down the region responsible for stable TGN/EE localisation of VHA-a1 compared to previous work, and to my knowledge this is the first description of a specific motif required for stable TGN/EE localisation in plants. The authors phylogenetically show that the TGN/EE-retention motif that they describe is limited to seed plants, and elegantly demonstrate that this region from other seed plants is capable of directing chimeric VHA-a1 to the TGN/EE in Arabidopsis, whereas this region from a non-seed plant is not. Expanding their work beyond Arabidopsis, the authors show that the single V-ATPase of Marchantia localises to both the TGN/EE and tonoplast in this liverwort, and suggest that this duality may be the “ancestral” state of V-ATPase localisation in land plants. Using CRISPR-Cas9 the authors surprisingly find that Arabidopsis *vha-a1* mutants have normal vegetative growth, which is in contrast to previous data using RNAi-mediated suppression. However, the authors later show that tonoplast-localised VHA-a2 and VHA-a3 appear to compensate for absent VHA-a1 in these mutant lines. Finally, the authors identify a specific requirement for VHA-a1 in pollen development that cannot be complemented by tonoplast-localised VHA-a2, VHA-a3 or the single Marchantia V-ATPase. This suggests a previously undescribed role for VHA-a1 in pollen development that has arisen in seed plants.

This manuscript is likely to be of high interest to the membrane trafficking community. It is generally well written, and the elegant model for V-ATPase trafficking presented in the conclusion places well into context both data presented here and previously published data. However, some of the data presented in this manuscript do not obviously support the conclusions made by the authors. This is likely due to presentation issues, and provided that the authors can amend this satisfactorily, this manuscript would – to our mind – be suitable for publication in *eLife*.

1) The current presentation and emphasis are adequate for a plant membrane trafficking audience; for a broader audience, however, a greater attempt could be made at extracting the deeper meaning and implications of the findings for protein sorting in general. This would do better justice to this contribution, which is important with respect to ER-export, TGN/EE retention and compartment acidification. Specifically, the Abstract is bogged down in detail and ends on a conclusion rather than on a perspective; the model for V-ATPase trafficking might be a more compelling final note for the Abstract. Similarly, the last paragraph of the Discussion, which announces itself as a conclusion, outlines rather focused, VHAa-specific questions.

2) The data as currently presented in Figure 2 (specifically Figure 2E, G, H, I and J) is not clear and does not immediately support the conclusions made in the accompanying text. The authors present this data as merged-channel images presumably to make concise use of space but this may have rendered some of the data unclear. For example, in Figure 2E the authors claim that the localisation of E156Q and VHA-a1: RFP are different (i.e. E156Q does not label the TGN/EE) and yet there is plenty of co-localisation evident in 2E. This may be due to VHA-a1 RFP signal overlying unrelated background from E156Q, but this is not easily evident from a merged-channel image. Moreover, in Figures 2G-J, the authors claim that the VHA-a1 variants presented label both the TGN/EE and tonoplast, but any TGN/EE signal from these variants is very difficult to see in these images. These data may become a lot clearer if the authors present the image channels separately. Data presented in Figure 4 supports the conclusions of Figure 2 much more clearly, but readers should not have to rely on a future figure to confirm this.

3) Subsection “The Marchantia V-ATPase is dual localized at the TGN/EE and tonoplast and is functional at the tonoplast in Arabidopsis” and Figure 8—figure supplement 1. The authors claim that MpVHA-a-mVenus labels both the tonoplast and punctae in Marchantia thalli that they then interpret to be dual TGN/EE and tonoplast localisation. However, we are really struggling to see “punctae” in these images labeled by MpVHA-a-mVenus.

4) Subsection “VHA-a1 has a unique and essential function during pollen development which cannot be fulfilled by VHA-a2, VHA-a3 or MpVHA-a” and Figure 9—figure supplement 2. The authors claim that UBQ10: MpVHA-a-mVenus in Arabidopsis labels both the TGN/EE and tonoplast. Whilst we agree that it does appear to label both punctae and the tonoplast in Arabidopsis, the authors provide no co-localisation with another marker or endocytic tracer to confirm TGN/EE localisation.

Reviewer #3:

The manuscript "The Arabidopsis V-ATPase is localized to the TGN/EE via a seed 1 plant specific motif and acts in a partially redundant manner with 2 the tonoplast enzyme" by Lupanga et al. describes a protein motif required for ER exit and TGN retention of VHA-a1 in Arabidopsis. This is conserved among seed plants. An exemplary non-seed plant VHA-a is partially functional in Arabidopsis.

The manuscript is, although quite technical in nature, well written and concise. It presents an in depth follow-up on the Dettmer et al., 2006 paper, and the evolutionary aspect adds general interest. Experimentally, I think certain aspects can be improved upon, which I outlined below.

1) There are several instances in the paper where a TGN localization is confirmed by co-localization with FM4-64 after prolonged BFA treatment. I find this not overly convincing. I think the authors should instead use co-localization with another marker, such, for example, immunolocalization of ECHIDNA protein. This would allow a more robust confirmation and quantification of TGN localization and would not require lengthy introgression of transgenic lines.

2) Figure 2. The authors should quantify the effects, for example by testing co-localization with FM4-64 punctae. Thus, the classes would become more clear.

3) Figure 7—figure supplement 1. The authors should provide some sort of quantification to make a stronger point about the varying degrees of rescue.

4) Figure 10—figure supplement 1. The authors should provide quantitative data on ConcA sensitivity.

5) Figure 3. The VHA-a1-GFP pattern after Sar1bGTP induction looks quite different in B and D, any explanation?

---

## [Author Response]

Reviewer #1:[…] This paper has a nice combination of mechanistic and evolutionary questions and the model they propose seems internally consistent. The manuscript is written clearly, but it would benefit from some additional quantitative analyses.1) Please conduct quantitative colocalization analysis of micrographs. Two freely available and well-documented plugins for Fiji/ImageJ are DiAna (Gilles et al. 2017, Methods) and JaCOP (Bolte and Cordelières 2006, J Microscopy).

We have conducted quantitative colocalization analyses for micrographs in Figure 2 using the JaCOP plugin in Fiji. We report Pearson’s correlation coefficients as well as Manders coefficients for each VHA-a1-GFP mutant.

2) Tonoplast morphology seems quite different in some of the VHAa1-TD point mutants (particularly ELE in Figure 4); please clarify how "tonoplast" signal was defined for the quantification in Figure 4B. Also, what is the magenta signal in Figure 4A? This is not clear from the labels or figure legend and it does not seem to display the same post-DEX behaviour as either VHAa1 or SAR1bGTP as presented in Figure 3. If it is VHAa1-RFP signal, the data in Figure 4B (and 5B) might be better presented as a decrease in colocalization between VHAa1 point mutants and VHAa1-RFP (or FM4-64 in Figure 5), corrected for fluorescence intensity, since this might be clearer than trying to define which pixels belong to the tonoplast.

The tonoplast was defined as all long membrane stretches. Points were manually selected in each cell to make line profiles. Fluorescence intensity was then measured along these line profiles. The maximum fluorescence intensity for each line profile was recorded.

The magenta signal is Sar1b-GTP and not VHA-a1-RFP. Care has been taken to present micrographs with the same size format and of cells of the same developmental stage. The labels and figure legend have been revised.

3) I cannot see any colocalization between MpVHA-a-mVenus and mRFP-MpSYP6A in Figure 8—figure supplement 1. Please either revise the discussion of these images, or provide more compelling evidence of this colocalization (better images, quantification, treatment with BFA, if this works in Marchantia?)

We have revised the discussion of the images. We present new pictures of MpVHA-a-mVenus in Marchantia that better illustrate that MpVHA-a has a clear tonoplast localization and also a punctate localization. We do not conclusively state that MpVHA-a-mVenus localizes to the TGN/EE.

4) It seems that the data presented in Figure 7C does not include any biological replication since only three points are plotted and the legend clearly indicates "n=3 technical replicates". Please perform additional biological replicates for this experiment.

Two additional biological replicates were performed for the cell sap pH (Figure 7C).

5) Are VHAa2 or VHAa3 localized to the TGN in vegetative tissue of the vhaa1 mutants? This would provide a second line of evidence to the authors' genetic results that VHAa2/VHAa3 can replace VHAa1 during vegetative growth and it might help to clarify the differences observed between VHAa1 RNAi lines and VHAa1 CRISPR mutants.

We have rigorously tried to detect VHA-a3 at the TGN/EE under normal conditions in wild type as well as in the *vha-a1* mutant root cells and we have failed. VHA-a3 is only detectable at the tonoplast in root cells when we treat with Concanamycin A and when post Golgi trafficking is blocked such as with the dominant negative Rab 5 mutant (Feng et al., 2017). We have utilized VHA-a3 tagged with GFP and mRFP and with both fluorophores, VHA-a3 is not detectable at the TGN/EE. It may be that with brighter fluorophores that are now available, VHA-a3 might be detected at the TGN/EE. However, it has to be considered that the amounts of VHA-a2/a3 that are passing through the TGN/EE under normal conditions are too small to be detected.

6) The model in Figure 10 illustrates a kind of competition between ER trafficking components for V-ATPases containing VHAa1 for TGN targeting via a COPII-dependent mechanism and VHAa3 for tonoplast targeting via a COPII-independent mechanism. Since VHAa2/a3 cannot compensate for loss of VHAa1 in pollen development, then this model predicts that overexpression of VHAa2/a3 could potentially cause a pollen development phenotype by displacing some VHAa1 at the TGN. Similarly, the model predicts reduced trafficking of VHAa1-TD point mutants (or VHAa3-a1-TD) to the tonoplast when VHAa2/a3 are overexpressed. Have these ever been observed (e.g. in VHAa3-GFP lines, or with VHAa3-GFP in the VHAa1 RNAi line)?

Overexpression experiments are not ideal because, other subunits need to be considered. VHA-a1/a2/a3 can only leave the ER in an assembled complex. Therefore, in order to achieve increased export of VHA-a1/a2/a3 out of the ER, the other V-ATPase subunits also have to be overexpressed in their correct stoichiometry. Considering the number of subunits involved, this is not feasible because silencing problems may occur

In addition, the expression of *VHA-a2* and *VHA-a3* is already high in the wild type, based on RNA expression data. The crux of the matter is not amounts of VHA-a1 verses VHA-a2/a3 but rather that their affinity for COPII machinery is different.

Reviewer #2:[…] This manuscript is likely to be of high interest to the membrane trafficking community. It is generally well written, and the elegant model for V-ATPase trafficking presented in the conclusion places well into context both data presented here and previously published data. However, some of the data presented in this manuscript do not obviously support the conclusions made by the authors. This is likely due to presentation issues, and provided that the authors can amend this satisfactorily, this manuscript would – to our mind – be suitable for publication in eLife.1) The current presentation and emphasis are adequate for a plant membrane trafficking audience; for a broader audience, however, a greater attempt could be made at extracting the deeper meaning and implications of the findings for protein sorting in general. This would do better justice to this contribution, which is important with respect to ER-export, TGN/EE retention and compartment acidification. Specifically, the Abstract is bogged down in detail and ends on a conclusion rather than on a perspective; the model for V-ATPase trafficking might be a more compelling final note for the Abstract. Similarly, the last paragraph of the Discussion, which announces itself as a conclusion, outlines rather focused, VHAa-specific questions.

We have revised the Abstract. The model for V-ATPase trafficking that we propose is emphasized. The knowledge gained on the evolution of differential targeting of the V-ATPase in eukaryotic cells is also highlighted.

2) The data as currently presented in Figure 2 (specifically Figure 2E, G, H, I and J) is not clear and does not immediately support the conclusions made in the accompanying text. The authors present this data as merged-channel images presumably to make concise use of space but this may have rendered some of the data unclear. For example, in Figure 2E the authors claim that the localisation of E156Q and VHA-a1: RFP are different (i.e. E156Q does not label the TGN/EE) and yet there is plenty of co-localisation evident in 2E. This may be due to VHA-a1 RFP signal overlying unrelated background from E156Q, but this is not easily evident from a merged-channel image. Moreover, in Figures 2G-J, the authors claim that the VHA-a1 variants presented label both the TGN/EE and tonoplast, but any TGN/EE signal from these variants is very difficult to see in these images. These data may become a lot clearer if the authors present the image channels separately. Data presented in Figure 4 supports the conclusions of Figure 2 much more clearly, but readers should not have to rely on a future figure to confirm this.

The image channels are presented separately for better clarity. We have also conducted quantitative colocalization analyses using the JaCOP plugin in Fiji. Pearson and Manders correlation coefficients are reported.

3) Subsection “The Marchantia V-ATPase is dual localized at the TGN/EE and tonoplast and is functional at the tonoplast in Arabidopsis” and Figure 8—figure supplement 1. The authors claim that MpVHA-a-mVenus labels both the tonoplast and punctae in Marchantia thalli that they then interpret to be dual TGN/EE and tonoplast localisation. However, we are really struggling to see “punctae” in these images labeled by MpVHA-a-mVenus.

We have revised the discussion of the images. We present new pictures of MpVHA-a-mVenus in Marchantia that better illustrate that MpVHA-a has a clear tonoplast localization and also a punctate localization. We do not conclusively state that MpVHA-a-mVenus localizes to the TGN/EE.

4) Subsection “VHA-a1 has a unique and essential function during pollen development which cannot be fulfilled by VHA-a2, VHA-a3 or MpVHA-a” and Figure 9—figure supplement 2. The authors claim that UBQ10: MpVHA-a-mVenus in Arabidopsis labels both the TGN/EE and tonoplast. Whilst we agree that it does appear to label both punctae and the tonoplast in Arabidopsis, the authors provide no co-localisation with another marker or endocytic tracer to confirm TGN/EE localisation.

We performed colocalization experiments with FM4-64 and BFA. The core of BFA compartments is labeled with MpVHA-a-mVenus and FM4-64 indicating that MpVHA-a is also present at the TGN/EE in *vha-a1*.

Reviewer #3:[…] The manuscript is, although quite technical in nature, well written and concise. It presents an in depth follow-up on the Dettmer et al., 2006 paper, and the evolutionary aspect adds general interest. Experimentally, I think certain aspects can be improved upon, which I outlined below.1) There are several instances in the paper where a TGN localization is confirmed by co-localization with FM4-64 after prolonged BFA treatment. I find this not overly convincing. I think the authors should instead use co-localization with another marker, such, for example, immunolocalization of ECHIDNA protein. This would allow a more robust confirmation and quantification of TGN localization and would not require lengthy introgression of transgenic lines.

FM4-64 is an endocytic dye that has been shown to label intracellular compartments including the early endosome which is the TGN/EE in plants (Dettmer et al., 2006 ; Viotti et al., 2010). Post-Golgi compartments aggregate around and inside the BFA compartment. Early endosomal compartments accumulate in the core of BFA compartments (Geldner et al., 2001; Grebe et al., 2003). Therefore, when our proteins of interest are found inside BFA compartments we can take this as confirmation for TGN/EE localization.

But in essence, the BFA treatments were done because the signal of individual TGN/EEs was barely above the detection limit. Thus, immunolocalization experiments with ECHIDNA protein would not be helpful.

2) Figure 2. The authors should quantify the effects, for example by testing co-localization with FM4-64 punctae. Thus, the classes would become more clear.

VHA-a1-mRFP is the reference marker for the TGN/EE. We have used the best possible marker for the experiment. The colocalization is now quantified using the JaCOP plugin. Pearson and Manders correlation coefficients are reported.

3) Figure 7—figure supplement 1. The authors should provide some sort of quantification to make a stronger point about the varying degrees of rescue.

Rosette areas have been quantified and the data is presented in a box plot with all points displayed.

4) Figure 10—figure supplement 1. The authors should provide quantitative data on ConcA sensitivity.

We have performed hypocotyl length measurements of etiolated seedlings grown in the presence of DMSO and ConcA. The data is presented in a box plot with all data points shown.

5) Figure 3. The VHA-a1-GFP pattern after Sar1bGTP induction looks quite different in B and D, any explanation?

Different developmental stage of the cells and different planes of the cells. We have taken new pictures that show cells from the same developmental stage.